# KSTAR: An algorithm to predict patient-specific kinase activities from phosphoproteomic data

Sam Crowl[1,3], Ben T. Jordan [1,3], Hamza Ahmed [1], Cynthia X. Ma [2] & Kristen M. Naegle [1]✉

Kinase inhibitors as targeted therapies have played an important role in improving cancer outcomes. However, there are still considerable challenges, such as resistance, non-response, patient stratification, polypharmacology, and identifying combination therapy where understanding a tumor kinase activity profile could be transformative. Here, we develop a graph- and statistics-based algorithm, called KSTAR, to convert phosphoproteomic measurements of cells and tissues into a kinase activity score that is generalizable and useful for clinical pipelines, requiring no quantification of the phosphorylation sites. In this work, we demonstrate that KSTAR reliably captures expected kinase activity differences across different tissues and stimulation contexts, allows for the direct comparison of samples from independent experiments, and is robust across a wide range of dataset sizes. Finally, we apply KSTAR to clinical breast cancer phosphoproteomic data and find that there is potential for kinase activity inference from KSTAR to complement the current clinical diagnosis of HER2 status in breast cancer patients.

Kinases make up the largest fraction of FDA-approved drugs for oncology[1], a reflection of their importance in oncogenesis and cancer progression. These drugs are also one of the most prevalent examples of precision medicine. For example, patients with BCR-ABL leukemias, HER2-overexpressing breast cancers, or EGFR-driven lung cancers benefit immensely from kinase-targeted therapies (often as adjuvants to chemotherapy, radiation, and/or surgery). Unfortunately, the selection and success of kinase inhibitors is often hampered by development of resistance mutations[2,3], failure to respond to treatment[4], and a limited set of current targets[1]— most new molecular entities target kinases for which inhibitors already exist. Hence, there is a need in the field of oncology to identify whether a patient might benefit from a kinase-inhibitor therapy and which kinase target(s) would be most beneficial.

In recent years, due to advances in proteomics, large-scale monitoring of protein phosphorylation is closer to being used within clinical profiling of tumor biopsies[4–8]. Since phosphorylation is a direct consequence of active kinases, or inactive phosphatases, this measurement might hold the key to better precision medicine. However, generating kinase activities from this data requires overcoming several challenges, including: (1) data sparsity or missing data—in shotgun phosphoproteomics (i.e., discovery-based approaches) lack of detection of a phosphorylation site may not be evidence that it is not present in the sample and although CPTAC approaches often use a reference standard, only 5 to 15% of all phosphorylation sites are common across patient cohorts, (2) there is an extreme paucity of data regarding the direct connection between phosphorylation sites and their kinases (only 5% of phosphorylation sites are annotated with a kinase)[9], and (3) challenges in relating quantitative data available from phosphoproteomics experiments to kinase catalytic activities. Quantification is particularly challenging—unless one uses a known spike-in for absolute quantification for every phosphopeptide of issue, all quantification

[1]University of Virginia, Department of Biomedical Engineering and the Center for Public Health Genomics, Charlottesville, VA 22903, USA. [2]Department of Medicine and Siteman Cancer Center, Washington University in St. Louis, St. Louis, MO 63108, USA. [3]These authors contributed equally: Sam Crowl, Ben T. Jordan. ✉e-mail: kmn4mj@virginia.edu

(label and label-free) is relative—i.e., it is not possible to understand differences in quantities between peptides, only differences of a peptide between conditions. This is due in large part to peptide-specific sample losses and ionization[10]. For example, if there are two kinases, each with a different substrate, and one substrate changes from 1fmol to 2fmol, but the second substrate changes from 8pmol to 16pmol, there is a 4000-fold difference in catalytic activity between the two kinases, but relative quantification interprets both of those as the same (2-fold different). Despite these challenges, phosphorylation is still more closely connected to kinase activity than commonly used proxies, such as mRNA, which rarely correlates with protein expression[11] or kinase activation loop phosphorylation, since kinase activity is regulated by a myriad of complex mechanisms[12–17].

Although there has been excellent progress in algorithm development to convert phosphoproteomic data into scores, rankings, or activity values for kinases, these algorithms often suffer from the major issues of phosphoproteomics (see Supplementary Table 1 for a summary). All but one algorithm we assessed, KEA3[18], depends heavily on the use of quantification of phosphorylation sites in a calculation for activity, via paired samples with relative quantification (PTM-SEA[19], KSEA[20], IKAP[21]), single sample normalized intensities (KARP[22]), or spectral counts (INKA[23]), despite the limitation that these values are not comparable across peptides or indicative of concentrations or substrate amounts. Most algorithms rely on sparse kinases-substrate information, most commonly from PhosphoSitePlus (KARP, IKAP, KSEA), where 95% of the phosphoproteome is unlabeled (no known kinase interaction). This results in the absence of kinase information for >80% of sites identified in a typical phosphoproteomic experiment (Supplementary Note 1). For those algorithms that turn to global kinase-substrate predictions, specifically NetworKIN[24], to expand the number of useable phosphorylation sites in an experiment (INKA and KSEA), NetworKIN predictions are used by applying a threshold to the weighted graph (edges indicating likelihood of a kinase-substrate interaction), where only edges above a certain score are kept. However, we recently showed that these thresholded graphs are highly problematic[25] and is likely the reason a systematic evaluation showed that NetworKIN-based annotations in KSEA performed worse than literature-based annotations[26].

In this work, we were explicitly seeking to create an algorithm that would be useful for patient care and: (1) can be used without requiring pooled or comparative samples (single sample), (2) avoids dependencies on the highly problematic nature of mass spectrometry-based quantification, (3) utilizes more of the phosphorylation data, but handles the issues with kinase-substrate predictions, and (4) avoids proxies of activity, like activation loop phosphorylation. Here, we present an algorithm that uses statistical and graph-theoretic approaches to infer the likely kinase activities from large-scale phosphoproteomic data. The underlying hypothesis is based on the action of kinases—given increasing activity, there will be an increasing number of observed substrates from a kinase's network. Ultimately our algorithm's kinase activity score is a reflection of "net kinase activity", since substrate phosphorylation is a reflection of the balance between kinase and phosphatase activity. In this work, we explain the details of our algorithmic approach and explore experiments to test whether the inferred kinase score changes with kinase activity. We demonstrate that we can predict: (1) increases in expected kinases as a result of network stimulation, (2) decreases in expected kinase activities as a result of kinase inhibition, and (3) tissue-specific kinase activity profiles, which are significantly more robust than the phosphoproteomic profiles, even across samples collected in different labs on different proteomic pipelines. Finally, we apply KSTAR to breast cancer biopsy-derived phosphoproteomic data and find that kinase activity profiles predicted by the algorithm can help identify misclassified HER2-positive breast cancer patients and identify clinically diagnosed HER2-negative patients that might respond to HER2-targeted therapy. Based

on these experiments and comparison to existing algorithms, KSTAR is a first-in-class algorithm that can utilize phosphorylation sites observed from single or multisample experiments and convert that to quantifiable, physiologically relevant, and interpretable insight into kinase activity with special strengths in accuracy in tyrosine kinase prediction and increased sensitivity to smaller numbers of phosphorylation sites and decreased reliance on a small subest of well-studied sites.

## Results

The KSTAR algorithm (Fig. 1) is based on the hypothesis that the more active a kinase is, the more of its substrates will be observed in a phosphoproteomic experiment. At its core, KSTAR is an algorithm that takes, as input, a set of phosphorylation sites observed in a mass spectrometry experiment, maps them onto global kinase substrate prediction graphs of KinPred[25], and converts the experimental input into a statistically robust KSTAR 'score' for each kinase, which increases with increasing representation of substrates from that kinase's network. When an experiment contains multiple conditions with relative quantification across all sites, the reported abundances are converted to binary evidence using a threshold relevant to the biological problem in question. We found that KSTAR predictions are fairly robust to changes in the threshold used as inclusion criteria for phosphorylation sites of an experiment (Supplementary Note 1).

### KSTAR algorithm

The first key development of KSTAR was to avoid the use of annotations alone, since so few phosphorylation sites are annotated, and to improve the usability of kinase-substrate predictions. As published and traditionally used (thresholded by removing edges below a cutoff score), predicted kinase-substrate networks suffer from the following issues: (1) the exclusion of a large number of phosphorylation sites from the human phosphoproteome, (2) high degrees of overlap between kinases, leading to a lack of discriminability for algorithms relying on these networks, and (3) high centrality of well-studied kinases and substrates (the more a substrate is annotated, the more kinases it is connected to)[25]. To overcome the challenges of prediction-based networks, we developed a "heuristic pruning" approach (Fig. 1) to create many possible, alternate representations of kinase-substrate relationships from global kinase-substrate prediction algorithms. In this work, all results are generated using the kinase-substrate graph of NetworKIN[24], with serine/threonine and tyrosine graphs being treated independently, since they are non-overlapping.

To achieve the objectives of pruning, we probabilistically select edges from a dense graph including all kinase-substrate predictions of the human proteome based on edge weights, where higher edge weights are more likely to be selected. This continues until all kinases are attached to a fixed number of substrates. The probabilistic selection introduces heuristic effects – creating different versions of output graphs each time the algorithm is run. To overcome the issues that happen when selecting only high edge weights in the graph (i.e., as done by thresholding), we apply constraints on the selection of edges to maintain specific properties of the output network. These constraints are: (1) the distribution of substrate "study bias"—as defined by the number of compendia substrates are documented in[25]—is the same for all kinases based on the distribution of the entire phosphoproteome in order to reduce kinase- and experiment-specific false positive rates (Supplementary Note 2) (2) edge selection is impossible once a substrate and/or kinase hits a maximum target in order to avoid the emergence of "hub substrates" and/or "hub kinases" and ensure each substrate only provides evidence for its most likely kinases, and (3) all kinases have the same number of edges in the final network to ensure that kinases are not isolated from the network, due to having only low probability edges. The example graphs shown in Fig. 1 depicts the success of this approach for reducing homologous kinase overlap – on

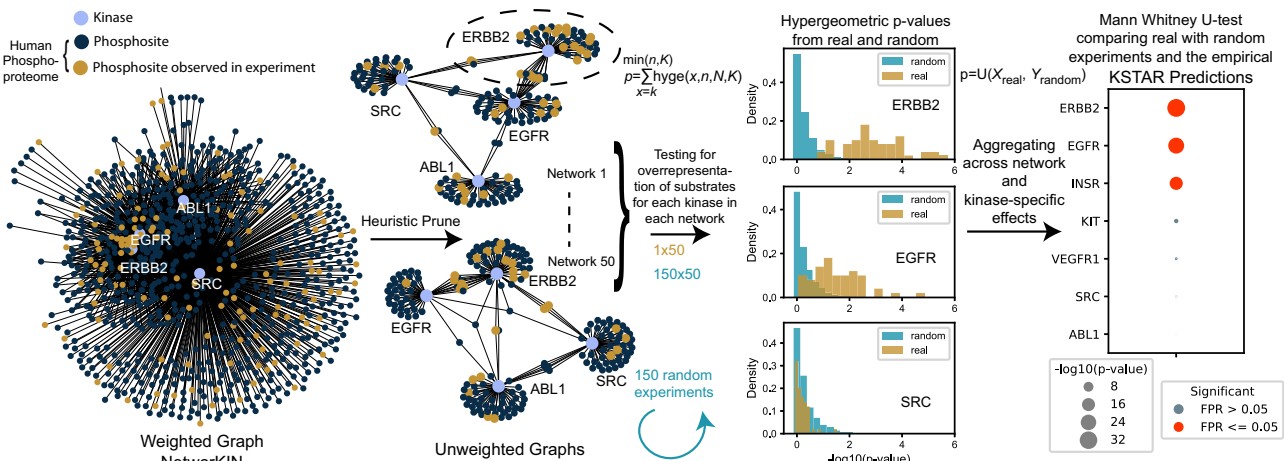

**Fig. 1 | Overview of KSTAR algorithm.** First, we heuristically prune dense and highly overlapping weighted kinase-substrate prediction graphs from NetworKIN[24] into many sparse, binary graphs. Statistical enrichment is calculated for an experiment that has a defined set of phosphorylation sites for every kinase across all networks using a hypergeometric distribution. We generate and calculate enrichment in 150 random experiments using the same approach. Next, we use the Mann-Whitney U test to measure the likelihood that the enrichment *p*-values in the real experiment are more significant than the random experiments, giving us a final *p*-value, which accounts for the underlying enrichment of substrates in a network, aggregates that information across the different network configurations, and controls for the kinase- and experiment-specific behavior of enrichment that occurs by random chance. We measure the false positive rate by measuring the distribution-based test for a random experiment against the remaining 149 random experiments, repeating this for 100 times. Finally, the numerical KSTAR "score" (the -log10 transformation of the Mann−Whitney U-test) is presented in graphical format where the dot size is larger when there is more evidence phosphorylation sites are coordinately sampled from a kinase network. The FPR is indicated by "Significance" of having less than a specific empirical FPR. Source data are provided with this paper.

average, KSTAR networks reduce the overlap of evidence between EGFR and ERBB2 for sites observed in an experiment from 80% in a thresholded network to 18% overlap. We have provided an extended description of the problems addressed by the prune procedure as well as pseudo-code for the generation of KSTAR networks in Supplementary Methods. Final pruned networks are then cast into unweighted graphs (i.e., binary graphs), which, relative to PhosphoSitePlus annotations and a thresholded NetworKIN graph, include more unique substrates, has lower overlap of evidence between kinases, and has better network coverage of experimentally identified sites (Supplementary Note 1).

The first statistical test in KSTAR measures overrepresentation of substrates in each kinase network, controlling for the likelihood that the observation happened by random chance. Since we use binary evidence (a substrate was observed) with binary networks, we can use a well-defined theoretical probability function to estimate this overrepresentation – the hypergeometric distribution, where we calculate the probability of having observed *k* or more substrates of a kinase in an experimental dataset with *n* phosphorylation sites, when the full graph (all sites in the human phosphoproteome) has *N* total phosphosites and a kinase has *K* total substrates. This approach has the benefit of scaling with the size of the dataset and the distribution of edges in the graph. The output of this calculation is a *p*-value for each graph (i.e., a set of 50 *p*-values for every kinase).

We next wished to control for the variability of the resulting hypergeometric p-values across the different network configurations and the likelihood of making the same observations in a random experiment exhibiting similar properties as the experiment under consideration. To do this, we create 150 random experiments, where we randomly draw the same number of phosphorylation sites as the real experiment from the background phosphoproteome, such that the distribution of substrate study bias is the same as the real experiment. For each random experiment, we calculate enrichment in each kinase network as we did for the real experiment, producing a "random" distribution of *p*-values. This random distribution can then be compared to the real distribution using the Mann Whitney U-test, a non-parametric, distribution-based test which estimates the probability that the ranked p-values of the real experiment are more significant than those of the random experiments. Fig. 1 shows examples of distributions that arise from this process. For EGFR, Fig. 1 indicates that some random networks with real data give low -log10 (*p*-values), yet the preponderance of evidence across all networks is more significant than what was observed in the random experiments, resulting in a small U-test *p*-value and a large KSTAR 'score' (the -log10(MW U-test *p*-value)). In contrast, it is clear that the p-values observed in SRC networks of the real experiment are no more significant than what occurs in random experiments, resulting in a small KSTAR score. Finally, we measure a false positive rate for the observed Mann Whitney *p*-value by measuring how often that *p*-value or more significant is observed in an empirical null model (repeatedly calculating the Mann Whitney p-values for treating a random experiment as the real experiment and comparing it to the remaining random experiments). The KSTAR score must be interpreted with the false positive rate (FPR) as we have observed some experiments result in large kinase-specific KSTAR scores that occur by random chance, due to the particular composition of the experiment. We detail the process by which we arrived at this approach to control and measure for kinase- and experiment-specific false positive rates in Supplementary Note 2, which also shows before these controls were added we saw very high false positive rates for kinases that demonstrate high study bias (e.g., 100% FPR for FYN, LCK, and HCK). Hence, the final KSTAR score is a statistically powered value that ultimately reflects the fundamental measurement of the proposed hypothesis – an activity score that increases when more evidence is observed across many different possible network architectures and is more significant than what is observed by random chance alone.

## KSTAR correctly infers expected tissue-specific kinases, kinase activation, and kinase inhibition

We wished to understand if KSTAR scores correlate with kinase activities and so we explored KSTAR predictions in-depth for specific physiological test cases of kinase activation and inhibition. Here, we summarize a set of key kinase predictions from KSTAR, with full activity predictions and their false positive rates presented

in Supplementary Note 3. We find that KSTAR predicts expected activity changes in response to either activation or inhibition and introduces interesting hypotheses. We also found from these experiments that using the base kinase-substrate prediction network from NetworKIN[24] results in predictions consistent with physiological expectation, but using kinase-substrate networks from other prediction algorithms (GPS[27] and PhosphoPICK[28]), did not result in physiologically relevant hypotheses. For example, PhosphoPICK-based networks failed to show ABL activity in BCR-ABL driven cancers and GPS failed to show HER2-activity in any breast cancer sample, where HER2-activity is a driving oncogene in many of these samples. Instead, GPS-based networks suggested that ERBB2 activity increases in Jurkat cells in response to TCR activation, inconsistent with tissue-specific expectations. Therefore, we have used NetworKIN as the foundation for KSTAR.

In our first study we explored growth factor stimulation in epithelial cells. We used two phosphoproteomic studies of a human mammary epithelial cell line (184A1) stimulated with EGF to explore whether KSTAR accurately predicts the onset of EGFR activation and other related EGFR-network kinases. One of the studies additionally includes a HER2-overexpressing model of 184A1 cells (24H cells, expressing 600,000 vs. 20,000 copies of HER2/ERBB2)[29] and measured response to EGF- and HRG-stimulation, where EGF drives EGFR:ERBB2 dimers and HRG drives ERBB2:ERBB3 dimers. Despite the different phosphoproteomic pipelines used in the two studies, the KSTAR predictions are similar for parental 184A1 cells stimulated with EGF (Supplementary Note 3 and Fig. 2a), suggesting no basal EGFR or ERBB2 activity exists after serum starvation and is followed by rapid onset and peak activation between 4 and 10 min post-stimulation. Notably, cytosolic kinases that are not expected to be in the epithelial lineage are not predicted to be active in any condition (Supplementary Note 3). In the HER2-perturbation experiment there are predictions of both dynamics and activity differences between cell lines and growth factors (Fig. 2a), consistent with the expected biology of the cells. For example, there is basal EGFR activity in HER2-overexpressing cells, slower dynamics of ERBB2 activation by HRG, and lower activity of both receptors from HRG-treatment – all of which are consistent with the effects of HER2-overexpression, the increased migration of 24H cells in the absence of stimulation, and the maximal migration upon EGF-addition[29]. Finally, and somewhat surprisingly, we see maximal statistical significance of ERBB2 in the EGF stimulation of both cell lines, with some indication that HER2-overexpression sustains that activity longer in the 24H cells. The failure to see additional increases in ERBB2 activity in HER2-overexpressing cells may be a result of our statistical limit, as no additional gains can be seen beyond this amount of evidence that is already present in the parental cells stimulated by EGF. Taken together, the specific kinases predicted active, their associated scores, and their patterns of activation suggest that KSTAR predictions are capable of reproducing expected biology and that the use of evidence in experiments alone (i.e., the removal of quantitative fold changes) is useful for making comparative hypotheses between conditions.

Next, in order to explore a cell lineage that is expected to have a distinctly different fingerprint of kinase activities than epithelial cells, we turned to datasets available in hematopoietic lineages, including several datasets in K562 (chronic myeloid leukemia) (Supplementary Note 5) and Jurkat cells (Supplementary Note 3). The Jurkat cell experiment by Chylek et al.[30] captured the fast dynamics of TCR activation at 5, 15, 30, and 60 s after stimulation. Importantly, the KSTAR predictions result in robust tissue-specific signaling expectations (Fig. 2b), predicting fast and robust (statistically saturated) activity of cytosolic kinases downstream of TCR, including LCK, FYN, HCK, BTK, and ITK (an important observation—our approach to controlling for high study bias of LCK, FYN, and HCK did not prevent them from giving robust activity values in a physiologically-relevant system). Slower to

reach maximal detectable activity include YES1, BLK, and FGR. Additional hypotheses suggested by KSTAR predictions include a slower (30-second) onset of the RTK activity of VEGFR2 and NTRK1. Most importantly, these tissue-specific kinases were not predicted to be active in the epithelial experiments and epidermal growth factor-specific kinases are not predicted to be active in Jurkat/TCR signaling. Hence, KSTAR predictions are consistent with tissue- and signaling-specific expectations.

Following exploration of stimulation experiments, we next wished to explore whether KSTAR could predict inhibition of kinases. The oncogenic BCR-ABL fusion protein drives some chronic myeloid leukemias (CML) and represents an important target for treatment. Unlike many kinase inhibitors, inhibition of BCR-ABL can initiate cell death pathways within the first hour of treatment. This occurs even though ABL activity has been shown to be recovered within 4–8 h after drug washout, depending on the study[31–34]. To validate the predicted tyrosine kinase activities from KSTAR, we compared the kinase activity profiles of the CML cell line K562 in response to treatment with the ABL-inhibitor dasatinib from a study by Asmussen et al., who profiled phosphopeptide abundance before treatment, at the time of drug washout (EOE), and 3 and 6 h post drug washout (HDP3 and HDP6, respectively)[34]. KSTAR predictions for ABL1 and ABL2 activity show a decrease following treatment, although the activity levels remain significant across all time points (Fig. 2c, Supplementary Note 3). Also, as expected, this activity is partially recovered after drug washout and the dynamic patterns of ABL kinases are mirrored in the Src family kinases (SFKs) BLK, HCK, and FGR, known alternate targets of dasatinib[2]. Finally, we note that there is robust down-regulation of a number of receptor tyrosine kinases, which is consistent with the conclusions of the original study[34] and others[32], which suggests dasatinib treatment in K562 cells is dependent upon on the elimination of ABL-GF-R interactions more than the elimination of ABL activity.

To this point, we have demonstrated the utility of KSTAR for predicting tyrosine kinase activities. To validate serine/threonine kinase predictions, we profiled the response of the breast cancer cell BT-474 to five different clinically relevant AKT inhibitors (Fig. 2d, Supplementary Note 3), based on data obtained from Wiechmann et al.[35]. BT-474 is a cell line that overexpresses ERBB2, which commonly leads to increased AKT activity. Upon applying KSTAR to this dataset, a complete elimination of AKT activity was predicted across all five inhibitors, suggesting that each inhibitor is highly effective at targeting AKT. While AKT3 was not identified by chemical proteomics in Wiechmann et al., our results suggest that AKT3 is active basally in BT-474 cells and that AKT3 activity is equally affected by treatment with these AKT inhibitors. We validated that the coordinated AKT1/2/3 predictions are not due to indiscriminate kinase networks—no AKT kinase shared more than 20% of substrates in KSTAR networks with any other AKT kinase. Hence, the basal activity predictions and predicted decrease of AKT family kinases in response to drug comes from independent networks.

Figure 2d highlights two interesting predictions from KSTAR. First, AKT inhibition appears to increase CSNK2A1 (Casein Kinase 2) activity in four of the five drugs. There is evidence that there is complex interconnections between CSNK2A1 and AKT kinases[36] and these predictions suggest that there might be an unintended increase in casein kinase activity as a result of AKT inhibition. Second, we observed interesting patterns in PRKACA/B, which shows decreasing activity with the ATP competitive inhibitors, but not the PH-domain binding allosteric inhibitor MK-2206. Hence, these predictions suggest the competitive ATP inhibitors bind to PRKACA/B kinases, and the greatest decreases predicted by KSTAR in PRKACA/B (GSK2110183, GSK690693, and AZD5363) were shown to bind PRKACA/B by chemoproteomics[35]. Hence, KSTAR predictions accurately predict AKT inhibition and help identify cross-talk at the network level (CK2) and the inhibitor level (PRKACA/B).

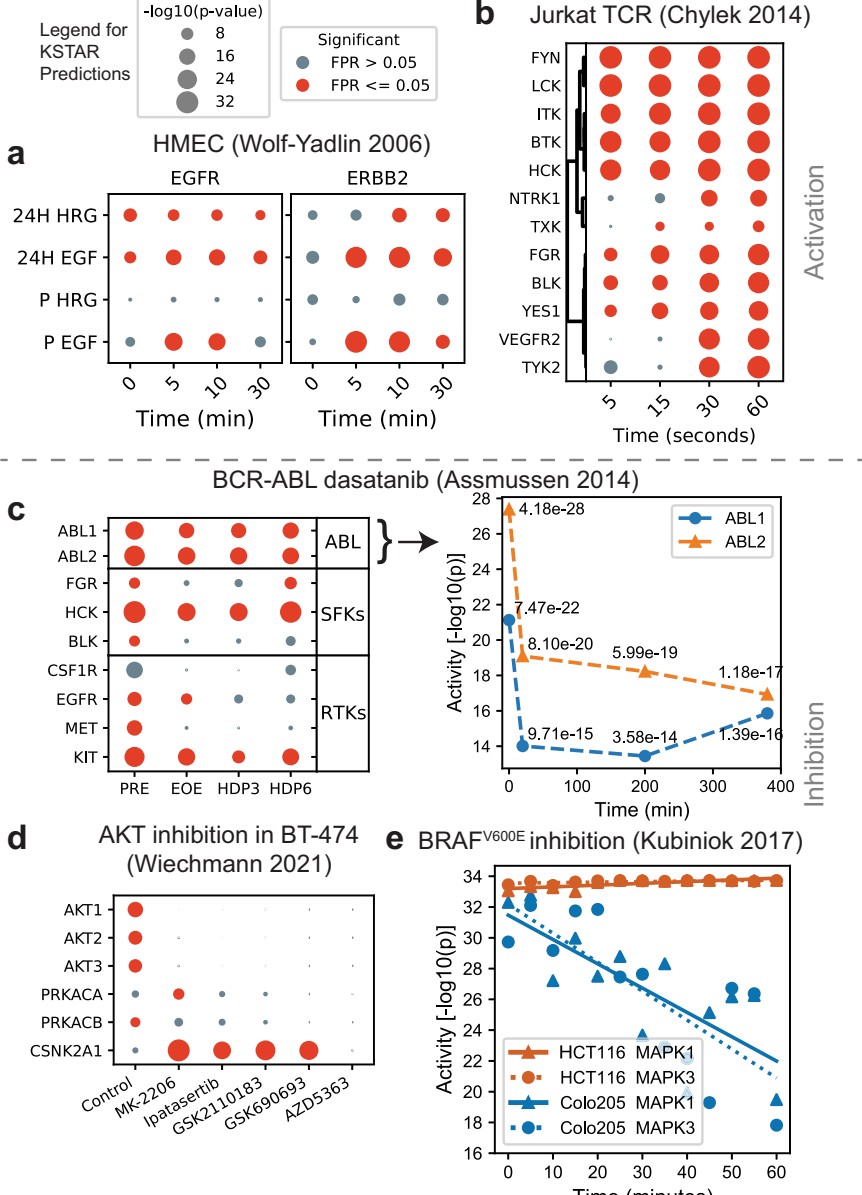

**Fig. 2 | KSTAR applied to diverse cell models of kinase activation and inhibition.** Full KSTAR results for data in this figure available in Supplementary Note 3). Panel titles give the reference for the publication study of the phosphoproteomic data. All KSTAR predictions use the same legend for score size and significance as given above panel A. **a** Predicted activation patterns of HMEC cell lines (P for parental 184A1 and 24H for HER2 overexpressing 184A1) in response to EGF and HRG stimulation. **b** Predicted activation patterns of TCR stimulation in Jurkat cells shows early and robust activation of TCR-specific kinases (this figure is in seconds). **c** Predicted kinase patterns in response to inhibition of BCR-ABL inhibition by dasatinib in K562 cells with a detailed plot of significance changes for the ABL

family kinases demonstrating a decrease, but continued activity of the oncogene. Kinase activity decreases in receptor tyrosine kinases (RTKs) correspond with findings of the original publication[34] as do changes in the off-target interactions with Src family kinases (SFKs). **d** AKT inhibition by five inhibitors, all competitive ATP inhibitors, except MK-2206 an allosteric inhibitor of AKT, demonstrate robust inhibition of all AKT homologs and interesting increases in CSNK2A1. **e** Vemurafenib treatment, targeting the BRAF[V600E] mutation found in Colo205 colorectal cancer cells, but not the HCT116 cell line, demonstrates a decrease in MAPK activity specific to BRAF mutation, although still statistically significant MAPK activity. Source data are provided with this paper.

While the previous studies help to demonstrate how KSTAR effectively predicts changes to kinases directly impacted by stimulation or inhibition, KSTAR is currently limited by the kinases that are present in NetworKIN and some commonly studied cancer-specific kinases do not currently have predictions, such as AXL, DDR2, and RAF. In order to determine whether KSTAR is still applicable in cases where direct kinase targets may be absent from predictions, we profiled the response of colorectal cancer cell lines to RAF inhibition by vemurafenib, using a time-resolved dataset generated by Kubiniok et al.[37] Vemurafenib is commonly used to target cells that harbor a BRAF[V600E]

mutation, such as the colorectal cancer cell line Colo205. However, it has achieved limited clinical utility, in part, because tumors that contain a RAS mutation or overactive RTKs often see an adverse response where the MAP-ERK pathway is activated rather than inhibited[37,38]. HCT116 cell line harbors a mutation in KRAS and provides an example of a cancer cell that exhibits this paradoxical response, which was also treated in the study by Kubiniok et al.

KSTAR predicts a clear decrease in MAPK1/3 activity over time in Colo205 cells, and a small increase in ERK activity is observed in HCT116, consistent with the paradoxical effect of RAS mutation in

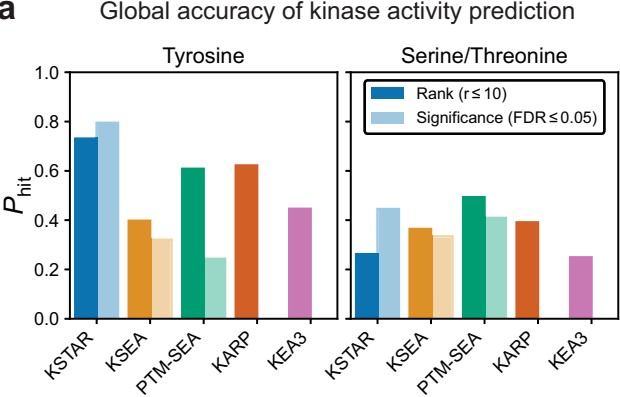

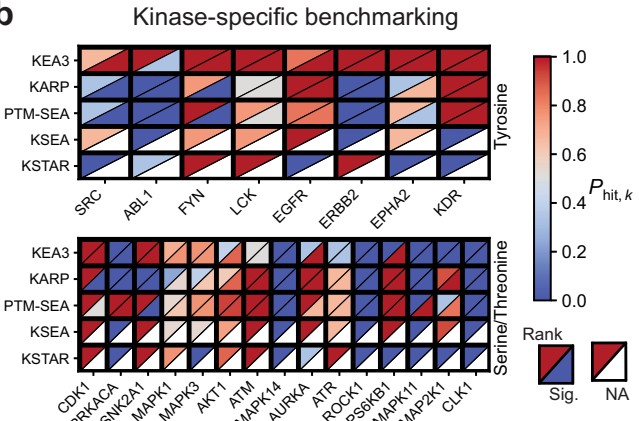

**Fig. 3 | Comparing accuracy of KSTAR to other available kinase activity algorithms.** KSTAR and four other publicly available kinase activity algorithms (KSEA, PTM-SEA, KARP, KEA3) were applied to a suite of inhibition and stimulation datasets. Accuracy measures expected activity changes as defined by $P_{hit}$—the fraction of conditions for which a perturbed kinase was found differentially active, either based on activity rank (in the top 10 kinases) or significance (FDR <= 0.05), which is not available (NA) for KARP and KEA3. **a** Global accuracy of each algorithm for tyrosine or serine/threonine kinases. **b** Kinase-specific accuracy of each kinase activity algorithm, separated based on accuracy metric (rank, upper left triangle or significance, lower right triangle) and kinase type (Tyrosine, Y or Serine/Threonine, ST). The heatmaps only include kinases for which all algorithms had available predictions (full heatmaps in Supplementary Note 4). Source data are provided with this paper.

HCT116 (Fig. 2e, Supplementary Note 3). However, the basal activity values limits the ability to observe significant increases in activity due to the statistical limit. The increase in activity becomes more apparent when the kinase activity profiles are transformed via quantile normalization across samples (Supplementary Note 3). Importantly, pre-treatment MAPK1/3 activity levels are comparable between HCT116 and Colo205, indicating that differences are a result of treatment and not basal activity. While the ultimate goal might be to expand NetworKIN and KSTAR to include these kinases in predictions, these results indicate that KSTAR is able accurately predict kinase activity profiles affected by missing kinases.

## Global accuracy benchmarking and comparison to other algorithms

Having observed that KSTAR activity scores are consistent with the detailed physiology of test cases, we next sought to globally compare KSTAR predictions to other available algorithms. Global benchmarking is difficult for three key reasons: (1) each algorithm is dependent on varying types of information such as relative intensities[22], differential fold-changes[19,39], or gene lists[18], (2) each algorithm produces very different types of outputs from ranks, to scores, or scores with accompanying significance values for a algorithm-specific number of kinases (see Supplementary Note 4), and (3) encoding globally positive and negative kinase sets for a given stimulation or inhibition condition is not necessarily biologically relevant or known, as kinase networks are inherently interconnected. For example, in the case of BCR-ABL driven CML cells, ABL inhibition is therapeutically effective by reducing RTK activation rather than causing a significant reduction in ABL activity. However, despite these complications, global benchmarking allows for a greater understanding of the general accuracy and best use cases of algorithms across a broad range of conditions and kinases. We adopted a similar benchmarking approach as Yilmaz et al.[40], in which we defined a hit as any instance in which a kinase expected to be perturbed was identified as differentially active, and a miss as any instance when the expected kinase is not identified as differentially active. We compiled multiple publicly available datasets representing a total of 15 studies with 51 total conditions allowing for the testing of 38 different serine/threonine (Supplementary Table 4) and 19 different tyrosine kinases (Supplementary Table 3). To be able to measure performance for the vast majority of algorithms, we used a rank-based

statistic—considering a kinase as a hit if it appears in the top ten most affected kinases. Additionally, for those algorithms that provide a binary active/inactive call based on statistical significance, we also measured a hit as whether the kinase was correctly considered as activated or inactivated in a differential condition. To avoid over/underestimation of accuracy in benchmarking as a result of multiple studies of the same kinase, we first measured the average accuracy for each tested kinase (heatmaps in Fig. 3b and Supplementary Note 4), then recorded the global accuracy as the average accuracy across all kinases (see methods for details). We compared KSTAR to the following four available algorithms that were compatible with benchmarking studies: KSEA[39], KARP[22], PTM-SEA[19], and KEA3[18].

Figure 3 highlights the average prediction accuracy for tyrosine and serine/threonine networks, both globally and for individual kinases. These results demonstrate that KSTAR consistently recovered tyrosine kinases expected to be perturbed. This performance is especially notable in cases where a binary call for activity is necessary, where KSTAR outperforms the next best algorithm by almost 50% and only fails for one kinase condition common to all algorithms (ABL). It is also notable that KARP, the only other algorithm that takes deliberate steps to account for study bias in kinase-substrate networks, exhibited the second best performance for tyrosine kinases based on rank (Fig. 3a). KSTAR's binary predictions for serine/threonine kinases also outperforms other algorithms, but to a lesser extent than tyrosine kinases. Interestingly, where KSTAR improves in performance as a binary predictor compared to rank-based measurement, KSEA and PTM-SEA significantly degrade in performance. KSTAR likely improves due to statistical saturation for certain kinases across most datasets, namely certain MAPKs and CDKs as is observed in the BRAF inhibition test case described in Fig. 2e, which makes rank-based performance difficult for other S/T kinases (for example, RPS6KB1 and AURKA were not found in the top ten, but were considered differentially active by their false positive rate). On the other hand, the reason for degradation of KSEA and PTM-SEA are likely two fold. First, kinases only have an associated activity score if at least one substrate of that kinase is identified in the sample, leading to variability in the number of kinases with predictions for each condition (Supplementary Note 4). This could lead to inflated rank-based accuracy, as it is easier to be in the top 10 of all kinases if there are only 20 other kinases (instead of 100) with predictions (a notable weakness of this benchmarking approach,

which does not account for total set size of kinases). Second, many kinase scores, particularly for tyrosine kinases, rely on only a few substrates, so if any of these substrates exhibit large fold-changes in the dataset, they will likely lead to large kinase scores. However, these scores are not robust, leading to low statistical significance. For this reason, the authors of both KSEA and PTM-SEA recommend restricting analysis to kinases with a set number of identified substrates, but this severely reduces the total number of kinases with predictions (Supplementary Note 4).

To assess whether the performance of other algorithms could be improved by expanding the number of kinase-substrate connections with the pruned networks developed in KSTAR, we applied KSEA to the benchmarking dataset using the pruned kinase-substrate networks generated for KSTAR (Supplementary Note 4). We found that over half of the individual pruned networks tested improved rank-based performance of KSEA compared to thresholded NetworKIN, but generally failed to outperform KSEA predictions generated from known kinase-substrate annotations from PhosphoSitePlus. However, rank-based accuracy for tyrosine kinases was best when aggregating information across all 50 pruned networks using the median activity scores, suggesting that pruned network ensembles could potentially improve the performance of other algorithms for tyrosine kinases. These same gains were not observed for serine/threonine kinases, though. Poor significance-based performance was observed for both tyrosine and serine/threonine kinases, highlighting the value of generating the random null distribution used in KSTAR to improve statistical robustness of predictions. While reformulation of other algorithms like KSEA for use in the KSTAR framework is intriguing, the use of quantification in these algorithms make the generation of a random null distribution that correctly reflects the study bias and quantification distribution of real experiments more difficult and beyond the scope of this work.

### Benchmarking sensitivity and study bias

In addition to globally testing recovery of specific kinases, we hypothesized that KSTAR's unique ability to use most of the phosphorylation sites identified in an experiment (Supplementary Note 1) would allow KSTAR to be more sensitive to identifying signal in smaller datasets. We were also interested in understanding if KSTAR predictions were less reliant on well-studied sites than other methods, which was a key goal of generating the pruned networks and well-controlled random experiments in Fig. 1. In addition to the observation that kinase annotations and predictions encode study bias[25], we found that phosphoproteomic studies are more likely to identify those sites of high study bias and that the more well-studied sites tended to exhibit larger fold-changes across the benchmarking dataset, particularly for tyrosine sites (Supplementary Note 2). Inspired by the idea of testing random versus targeted attacks on networks by Albert et al.[41], we developed an experiment to assess changes to the predicted activity of a kinase as data is removed, either by random loss of phosphorylation sites from an experiment or by targeted removal of the most well-studied sites, defined by the number of compendia a site is recorded in. If losses of both types are equivalent, it suggests low dependency on study bias, whereas a faster change in prediction significance for targeted loss suggests a high dependency on specific phosphorylation sites (Fig. 4b). We defined the tolerable loss of an algorithm as the maximum percent of sites that can be removed from a dataset before the majority of experiment replicates no longer indicate significant activity ($FDR \leq 0.05$). To measure global differences between the random or targeted removal of sites, we also defined an algorithm's "sensitivity to data loss" as the area under the random attack curve and an algorithm's "sensitivity to study bias" as the area under the curve between the random and targeted removal of sites (Fig. 4a).

We performed our network loss experiment across all conditions for which the perturbed kinase was found to have significant activity

when the full dataset was used ($FDR \leq 0.05$), comparing all algorithms that give a defined statistical call for kinase activity: KSTAR, KSEA, and PTM-SEA. We found that KSTAR predictions were remarkably stable, with most conditions tolerating >60% data loss and some conditions able to maintain significant predictions using only 5% of the data (Fig. 4c). While there is a decrease in the median tolerable data loss between the random and targeted removal approaches for tyrosine kinases (75 to 65%), these differences were not found to be statistically significant, suggesting that KSTAR is not heavily reliant on well-studied sites. On the other hand, both tyrosine and serine/threonine kinase predictions from KSEA, as well as serine/threonine kinase predictions from PTM-SEA, exhibited significantly less tolerable loss under targeted attack, demonstrating a high dependency on a few well-studied sites. A measure of study bias for PTM-SEA tyrosine kinase predictions is not attainable, since performance degraded significantly under random loss (17.5%).

To better understand global differences between how each algorithm handles random and targeted losses, we next assessed the cumulative sensitivity of each algorithm (areas under or between the curves from 0–50% data loss, instead of the intersections of the curves with the line of significance). Similar to what was observed in Fig. 4c, we found that KSTAR was significantly less sensitive to both data loss and study bias than KSEA and PTM-SEA for both tyrosine and serine/threonine networks (Fig. 4d). PTM-SEA and KSEA were particularly sensitive to these losses in tyrosine kinase networks. In many cases, we found that predictions could often not be generated by KSEA or PTM-SEA when 50% of the data was removed via the targeted approach due to the loss of all known substrates of a kinase (Supplementary Note 4). These experiments suggest that since KSTAR is built to retain and integrate across more information from a phosphoproteomic dataset, it is more robust for making inferences on smaller dataset sizes and is significantly less dependent on high-study bias sites than KSEA and PTM-SEA.

### Kinase activity profiles are more robust than phosphoproteomic data

Due to the high complexity and stochastic nature of mass spectrometry measurements, obtaining reproducible results across studies and across laboratories can often be difficult. Multiple inter-laboratory studies have demonstrated that differences in instrumentation and pipelines can greatly impact the peptides identified in a single run, with reproducibility being highest between technical replicates from the same instrument[42,43]. In addition, the choice of phosphopeptide enrichment method can further increase variability and lead to differences in the type of phosphopeptides identified[44]. Here, we sought to determine whether kinase activity profiles obtained from KSTAR can robustly identify similarities and differences between samples, even in cases where there is low overlap in the phosphopeptides identified by mass spectrometry. To do so, we obtained a total of 11 different phosphotyrosine datasets from 7 different studies and 5 different labs. Of these 11 datasets, 7 profiled the phosphoproteome of non-small cell lung carcinoma (NSCLC) cell lines with activating mutations in EGFR (H3255 and HCC827)[23,45–48] and 4 datasets profiled K562 cells, a chronic myeloid leukemia (CML) cell line containing the BCR-ABL fusion protein[23,34,44].

Based on the Jaccard similarity between sites identified in each dataset, most experiments exhibit low overlap with the other experiments (Fig. 5). The datasets with the highest overlap all stem from either the same study, the same lab, or both. In the only case where both K562 cells and NSCLC cells were profiled by the same study (dataset 6 and 10[23]), the NSCLC sample shared the highest site similarity with the corresponding K562 sample rather than NSCLC samples from other studies. Overall, it is difficult to identify a clear pattern of separation between the NSCLC cell lines and K562 cells with the phosphoproteomic datasets alone. However, kinase activity profiles

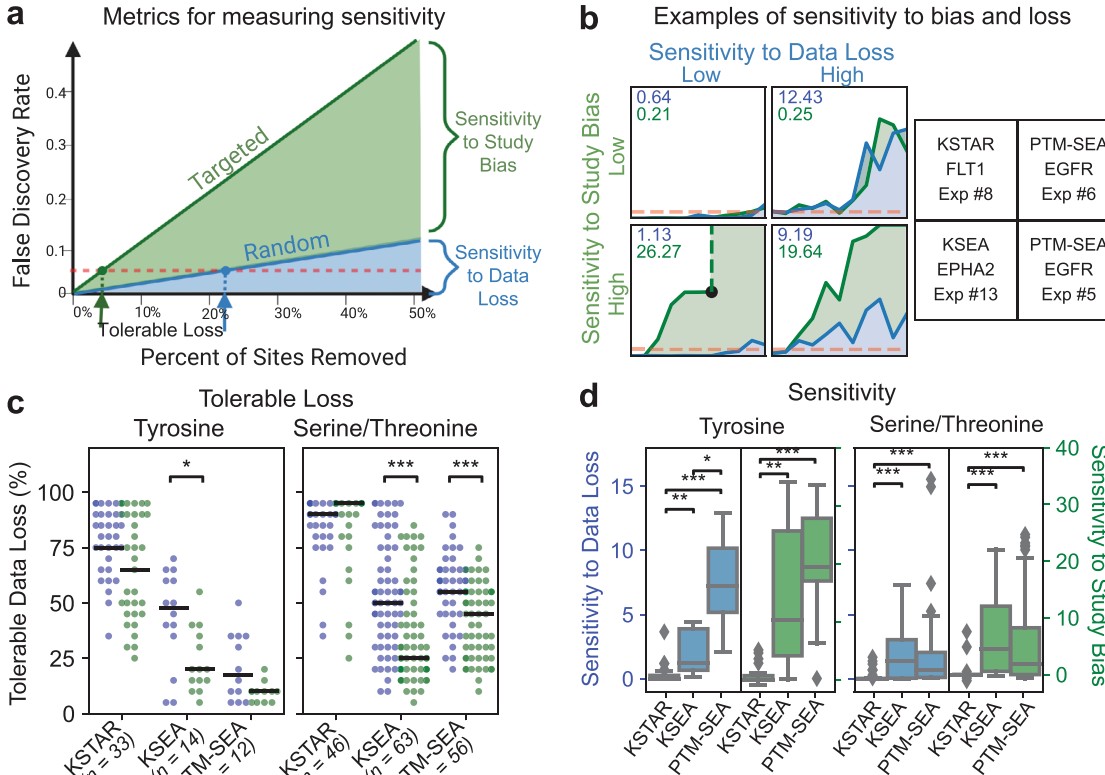

**Fig. 4 | Comparing sensitivity to data loss and study bias. a** Metrics defined to measure sensitivity to data loss and study bias. For a kinase in a prediction that starts as significant, we select data to be removed based on completely random selection or semirandom selection where high study bias sites are removed first. Results were obtained at every 5% loss increment, with each data point in the curve indicating the average false discovery rate across five replicates. Tolerable loss is defined as the percent of sites that can be removed before the majority of trials (3 out of 5) stop showing statistically significant activity for the kinase. Sensitivity is defined as the area under the random curve (data loss) or between the targeted and random curve (study bias). **b** Example loss curves that illustrate the difference between low or high sensitivity to data loss and/or study bias. The sensitivity to data loss (blue) and sensitivity to study bias (green) for each curve are displayed in the upper left of each plot. The right panels define the algorithm, kinase, and the benchmark experiment number (indicated in Supplementary Table 3 and Supplementary Table 4) that gave rise to these curves. The black dot in KSEA/EPHA2 in lower left quadrant indicates that KSEA was no longer able to calculate EPHA2 activity at that value of targeted data loss. **c** Tolerable loss under random (blue) or targeted (green) removal for all tested conditions for each algorithm (each dot represents the measurement of tolerable loss for a single condition, black line

indicates the median). Results are provided for tyrosine kinases (left) and serine/threonine kinase (right). Total number of conditions tested are given under the algorithm name. Only conditions where the perturbed kinase had statistically significant activity with the full dataset were used. To determine if the observed decrease in tolerable loss obtained between random and targeted attacks was statistically significant, a one-tailed Mann-Whitney U-test was used (*$p = 0.0099$, ***$p < 0.0001$). **d** The global measure, based on algorithm, for sensitivity to data loss (blue and left panels) or study bias (green and right panels). Box indicates median (center line), 25th and 75th percentiles (box boundaries), 1.5x the IQR of the box edge (whiskers), and any outliers beyond 1.5x IQR (points). If no outliers exist, whiskers indicate maxima or minima. Statistical significance was obtained from a two-tailed Mann–Whitney U test (*$p = 0.00007$, **$p < 1e − 5$, ***$p < 1e − 10$). A subset of biologically independent experiments from the benchmarking dataset in Fig. 3 were used for each algorithm, based on whether the perturbed kinase was predicted to have statistically significant activity ($FDR ≤ 0.05$) when the complete experiment was used (KSTAR (Y): $n = 33$, KSTAR (ST): $n = 46$, KSEA (Y): $n = 14$, KSEA (ST): $n = 12$, PTM-SEA (Y): $n = 12$, PTM-SEA (ST): $n = 56$). Source data are provided with this paper.

are capable of identifying shared profiles amongst all cells of the same type and reducing perceived similarity between studies from different cell types—datasets from the same cancer type were either moderately or strongly correlated ($r ≥ 0.47$, $p < 0.05$), while those from different cancer types were either uncorrelated or weakly negatively correlated ($r < 0.15$). When the same analysis was performed using KEA3[18], the only other algorithm capable of generating predictions in non-quantitative settings, kinase rankings tended to be highly correlated across all datasets ($r ≥ 0.68$, $p ≤ 0.05$), regardless of tissue similarity (Fig. 5d, Supplementary Note 5). This approach also represents a type of benchmarking that overcomes the reliance on a priori assumptions of kinase alterations—it only assumes that kinase profiles within a tissue should be similar and between tissues should be dissimilar. Hence, the transfer of phosphoproteomic data into kinase activity profiles using KSTAR greatly improves the comparability of independent phosphoproteomic experiments, a feat not currently attainable by any currently available algorithm.

To verify that the kinase profiles that improved similarity within cell types and discrimination between cell types was connected to the underlying biology, we explored the top-ranked kinases across each cell type (Fig. 5b). On average, EGFR was ranked as the most active kinase across all of the NSCLC cell lines, which is consistent with these lines carrying activating EGFR mutations. In K562 cells, the most active kinases were hematopoietic kinases HCK and BTK. Only MET appears as a highly active kinase in both cell types. We found that using only these top-ranked kinases was sufficient to separate the cancer types by hierarchical clustering (Fig. 5c). Further, the two NSCLC cell lines tended to cluster separately with this subset of kinases, which was not observed when using the entire kinase activity profile (Supplementary Note 5). Overall, these results suggest that KSTAR is able to robustly predict kinase activity profiles that define a particular tissue type and results in robust identification of samples coming from similar tissues, which is not possible from phosphoproteomics directly.

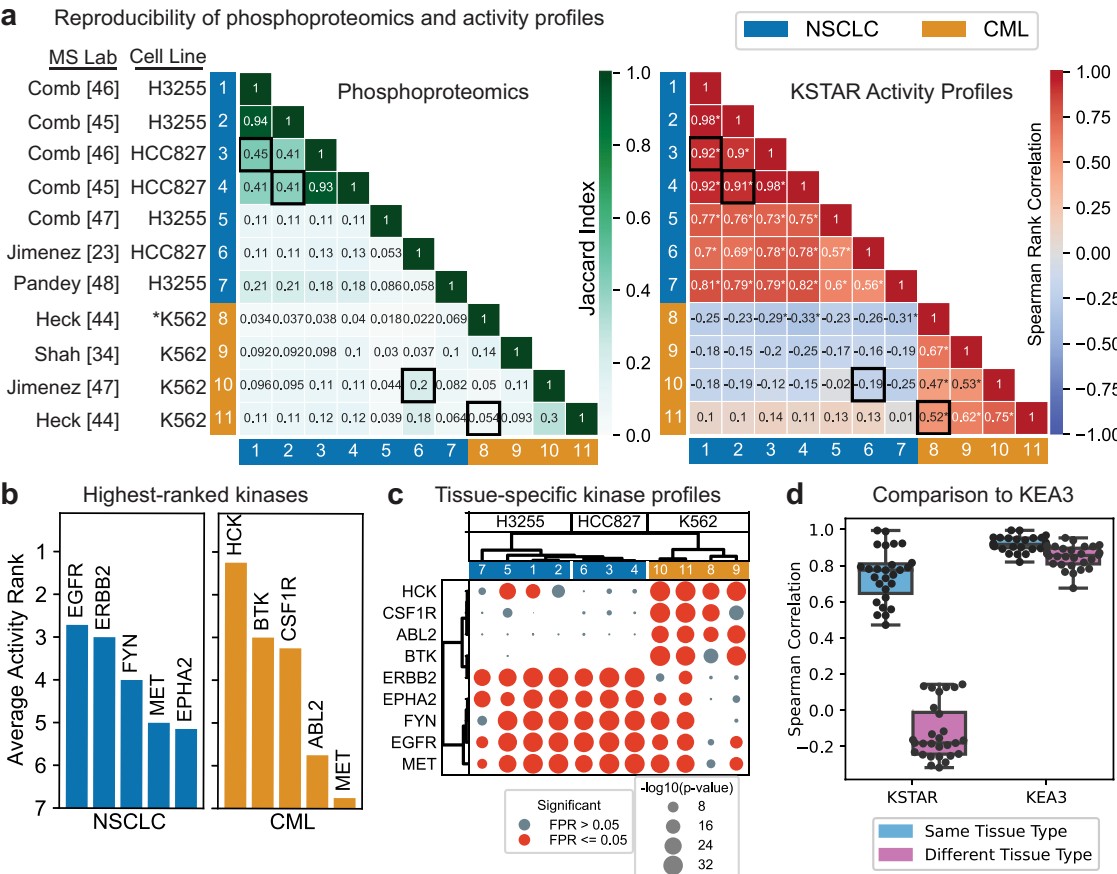

**Fig. 5 | Tissue-specific profiles of kinase activities across independent studies of non-small cell lung carcinoma (NSCLC) and chronic myeloid leukemia (CML) cell lines. a** Comparison of the phosphoproteomic results obtained by each study (left) and the kinase activity profile predicted by KSTAR (right). We used similarity metrics to match the different data types -- Jaccard similarity for phosphoproteomics and Spearman's rank correlation for kinase activity profiles. The ordering of the experiments in each heatmap is based on hierarchical clustering of the full kinase activity profile (Supplementary Note 5). **b** Kinases with the highest average activity ranking in NSCLC and CML cell lines. A rank of 1 indicates the most active kinase and a rank of 50 indicates the least active. For each study, kinases were sorted by their Mann-Whitney *p*-values to obtain the experiment-specific ranking, and then the average rank across experiments was calculated for each kinase. **c** Kinase activity profiles for top-ranked kinases. Both the kinases and experiments were sorted using hierarchical clustering with ward linkage. Full KSTAR results for data in this figure available in Supplementary Note 5. **d** A systematic evaluation of how KSTAR and KEA3 perform at identifying similarities between tissues of the same type and differentiate between tissues of different types based on predicted kinase activity/enrichment. KSTAR activity scores (or KEA3 kinase rankings) from each dataset were compared using Spearman's rank correlation, and results are plotted for within-tissue comparisons (NSCLC vs. NCLSC or CML vs. CML, *n* = 27 total pairwise comparisons across 11 biologically independent experiments) and between tissue comparisons (NSCLC vs. CML, *n* = 28 total pairwise comparisons across 11 biologically independent experiments). Box indicates median (center line), 25th and 75th percentiles (box boundaries), and the maxima and minima (whiskers). Points indicate a single pairwise comparison between experiments. Source data are provided with this paper.

## Kinase predictions in breast cancer

We next wished to ask if KSTAR predictions from bulk biopsy of human tumors might be informative for identifying patient-specific kinase activity profiles. We selected breast cancer in order to compare HER2/ERBB2 activity with clinical diagnosis of HER2-overexpression. We applied KSTAR to the CPTAC dataset of 77 breast cancer patients, using data from the consortium as published in Mertins et al.[5] that passed quality control for phosphoproteomics. Figure 6a focuses on ERBB2-activity predictions and the clinical status of the patient tumor. We considered three different cutoffs for making a binary decision of whether KSTAR predicts a tumor is "ERBB2 active", based on FPR (less than 0.05 and 0.1) or by the activity score (greater than 3, or having less than 1 in a 1000 chance that the number of sites observed in ERBB2 networks occurred by random chance). These different cutoffs produced varying rates of "true positives" (HER2+ patients predicted as having ERBB2 activity) from 27 to 50% and the selection of cutoff had a minimum impact on the false negative rate (19–24% HER2-negative tumors are considered active). All ranges showed excellent true negative rates (76 to 81% of HER2-negative tumors are predicted as

inactive). Since there are different ways to convert KSTAR scores into a binary prediction, we tested whether KSTAR rankings of ERBB2 activity was coordinated with HER2-status. Using a GSEA-style analysis to measure coordination between the continuous value KSTAR activity score and the binary label of HER2-status, we found significant coordination (*p* = 0.023). Unsurprisingly, due to the poor performance of other algorithms for tyrosine kinases and ERBB2 in particular in benchmarking, no other algorithms appear to be capable of predicting ERBB2 activity in a way that was correlated with HER2-status (Supplementary Note 6).

To our surprise, despite the fact that KSTAR predictions are well-correlated with HER2 status, some of the most ERBB2-active predictions occur in HER2-negative patients. Also, some HER2-positive patients demonstrate very low levels of ERBB2-activity. This difference between HER2-status and ERBB2-activity is feasible, since overexpression of HER2 may not necessarily lead to functional ERBB2 receptors at the cell surface and all breast epithelial cells contain some level of ERBB2, which could have activity without the requirement of amplification or overexpression. However, from this data, which lacks

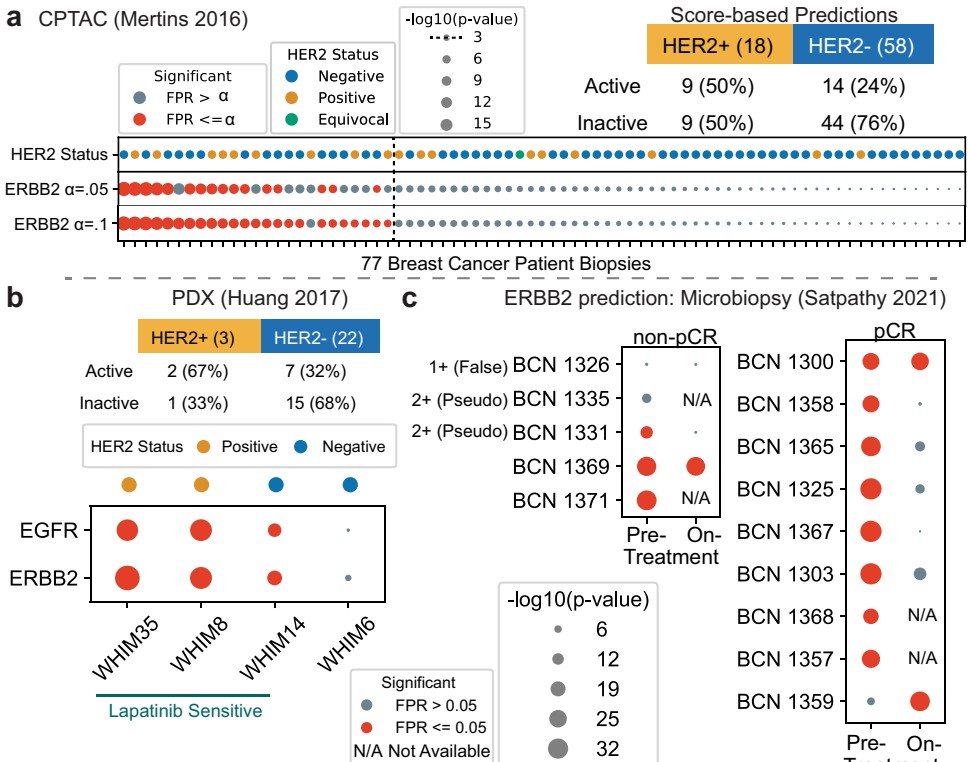

**Fig. 6 | KSTAR applied to breast cancer biopsies in three studies.** HER2 is used when referring to clinical diagnosis and ERBB2-activity for ERBB2/HER2 activity predictions. **a** KSTAR predictions of ERBB2-activity for the 77 breast cancer patients in the CPTAC dataset[5] and their clinical IHC/FISH HER2-status is given (samples are ranked by ERBB2-activity prediction score). The table gives the total number of HER2-positive and HER-negative patients and the KSTAR predictions for ERBB2/HER2 activity for the best of three cutoffs (score-based) considered for designation of ERBB2-active: *FPR* < = 0.05, *FPR* < = 0.1, and *score* >3. **b** Predictions of EGFR and ERBB2 activities for the patient-derived xenograft (PDX) models published in Huang et al.[8] subset that were treated with lapatinib (EGFR/HER2 targeted therapy), where WHIM14 is a HER2-negative tumor that showed a surprising response to

lapatinib treatment. The table reports the HER2-status of all 25 PDX tumors and KSTAR ERBB2/HER2-activity predictions. **c** The ERBB2-activity predictions for tumor biopsies of patients enrolled in a HER2-positive study by Satpathy et al.[4]. Five patients were non-pathological complete responders (non-pCR) and the remainder were pathologically complete responders (pCR). Biopsies were taken pre-treatment and most patient's also had an on-treatment biopsy taken with phosphoproteomic profiling. The first three non-responders were reclassified for HER2-status upon additional analysis in Satpathy et al. and results are shown as one false positive and two classified as "Pseudo-positives". Full KSTAR results for data in this figure available in Supplementary Note 6. Source data are provided with this paper.

longitudinal patient information, it is not possible to know if KSTAR predictions are accurately identifying the HER2-positive patients that will not be responsive to HER2-targeted therapies, a number that varies, but is thought to range between 35 and 50% of HER2-positive patients[4,49], or instead demonstrates issues with inferring tumor activity from complex bulk biopsies.

In order to test whether the departure of ERBB2-activity from HER2-status is clinically predictive of ERBB2-therapy response, we sought out additional datasets. These additional datasets include the phosphoproteomics of patient-derived xenografts (PDX) of breast cancer tumors[8], predominantly from HER2-negative patients, and the phosphoproteomics of HER2-positive breast cancer patients with data both before and 48–72 h after starting a combined chemotherapy and HER2-targeted therapy[4]. Although small, each of these datasets included response data to HER2-targeted therapy, allowing us to query whether KSTAR predictions are correlated with clinical response.

The PDX dataset[8] measured the phosphoproteomics of 25 breast cancer PDX models, which were predominantly HER2-negative. Of the three HER2-positive PDX models, we predict two of them are ERBB2-active and represent the highest activity levels of HER2/ERBB2 that we see across the entire dataset. We predict the third HER2-positive model as having globally low tyrosine kinase activity, including insignificant levels of ERBB2-activity. On the other hand, we predict 31.8% of the HER2-negative PDX models have evidence of ERBB2-activity, double the rate we observed in the CPTAC predictions. The researchers in this

study selected 4 PDX models to treat with the EGFR/HER2 inhibitor lapatinib. They selected two HER2-positive models, WHIM35 and WHIM8 (both of which we predict are the models we predict are ERBB2-active), and two HER2-negative models, WHIM14 and WHIM6. The authors were surprised to find that WHIM14 demonstrated significant response to lapatinib treatment. However, KSTAR predicts that the tumor has significant levels of both EGFR and ERBB2 activity, similar to WHIM35 and WHIM8, which also respond to lapatinib treatment. Hence, KSTAR predictions of EGFR and ERBB2 basal activity correspond with lapatinib response, including the surprising result of a HER2-negative tumor responding to HER2-specific therapy. This data suggests that kinase activity predictions from phosphoproteomic data may be capable of identifying patients who could benefit from HER2-targeted therapy despite being diagnosed in current clinical standards as HER2-negative. Together with the the CPTAC analysis, it suggests there are around 24 to 32% of HER2-negative patients that might benefit from HER2-targeted therapy.

Having explored a study that gave insight to the validity of HER2-negative patients with KSTAR-predicted activity, we wished to find insight into the set of HER2-positive patients and consistency with ERBB2-activity predictions. Satpathy et al.[4] focused entirely on HER2-positive patients and they developed a method to section a biopsy for a suite of molecular analyses, including microscaled proteomics and phosphoproteomics. Microscaling the analyses allowed the researchers the ability to perform replicates and analyses on biopsies taken 48

to 72 h following the commencement of a combination of chemotherapy and HER2-targeted therapy. Thirteen of the patients were treated with docitaxel and the combination of trastuzumab and pertuzumab, while one patient (BCN 1369) received paclitaxel and trastuzumab alone. In the study, 9 of the 14 patients were diagnosed as pathologically complete responders (pCR) and 5 patients were non-pathologically complete responders (non-pCR). This study provides an excellent opportunity to explore the reproducibility of KSTAR predictions across replicates, the relationship between therapy response and predictions of kinase activity pre-treatment, and the connection between response and ERBB2-activity, including on-treatment biopsy data, although complicated by the combination of treatment with chemotherapy.

From kinase activity predictions, we focused on the prediction of ERBB2 activity across replicates and samples, where we found consistent predictions across all replicates of a patient sample (Supplementary Note 6), which confirms that both the replication of the data and predictions from that data are robust. We saw a larger range of KSTAR activity scores in this dataset for ERBB2, compared to the TCGA/CPTAC dataset, likely due to a larger sampling of pTyr sites. Of all 14 patients that have been clinically diagnosed as HER2-positive, we predict 3 of the patients (BCN1326, BCN 1335, and BCN 1359) lack evidence of ERBB2-activity in the pre-treatment phosphoproteomic data, and an additional patient, BCN1331 with the next lowest activity and an FPR rate right at significance of 0.05. Three of these patients are non-pCR, suggesting that lack of ERBB2-activity basally may be a reason for non-response to chemotherapy combined with targeted HER2-therapy (Fig. 6c). In the original study, the researchers re-analyzed patient data for the non-pCR patients and found that BCN1326 was a HER2 false-positive, having no indication of ERBB2-amplification or increased protein expression and it is also the patient we predict has the lowest ERBB2-activity. Patients BCN1331 and BCN1335, also with low KSTAR scores for ERBB2-activity, were re-labeled by the research team as "pseudo-HER2-positive" having evidence of HER2 copy number increases, which did not translate to the protein levels. When KEA3 was applied to this same dataset, ERBB2 was not found to be higher than the 6th ranked kinase in any patient sample (Supplementary Note 6), whereas KSTAR predicted ERBB2 in the top 4 most active kinases in all the true basally HER2-positive patient samples, with the exception BCN1359. These results demonstrate that KSTAR predictions of phosphoproteomic data can complement clinical diagnosis of HER2 status, a designation that is clearly not perfect by IHC and FISH alone[4],[50].

KSTAR predictions from on-treatment samples are also promising, suggesting that six of the seven patients with pre-treatment ERBB2-activity are predicted to demonstrate a decrease, below significance, of ERBB2-activity on-treatment (Fig. 6c). There are two patients where predictions suggest there is no response to ERBB2-targeted therapy, BCN 1369 (non-pCR) and BCN 1300 (pCR). These data suggest that these patients are not benefiting from the HER2-therapy arm. Additionally, these two patients are most similar according to their tyrosine kinase activity predictions (Supplementary Note 6), suggesting that resistance to HER2-therapy might be encoded by their particular pattern of tyrosine kinase activities. BCN 1371, the last non-pCR patient, which has no on-treatment sample, has a similar kinase activity profile to the HER2- non-responders BCN 1369 and BCN 1300. Similarly, BCN 1357 and BCN 1368, which have no on-treatment phosphoproteomic data, cluster most similarly with BCN 1358 and BCN 1368, respectively, both of which show robust decreases in ERBB2-activity on-treatment. Where these similarities between pre-treatment patient responses and clinical outcomes are seen in tyrosine kinase activity profiles, they are do not occur in serine/threonine kinase activity profiles (Supplementary Note 6).

The final ERBB2-predicted negative patient, BCN1359, represents a unique case, as we predict it has low pre-treatment activity that

increases on therapy. Satpathy et al. did not redefine the HER2-status of this patient and their analyses show this patient also uniquely undergoes an increase in ERBB2 mRNA and protein upon treatment. A unique trait of BCN1359's pre-treatment kinase profile is EGFR-activity, in the absence of ERBB2-activity. This status is rare as we have typically observed, across the breast epithelial and breast cancer samples we have profiled, that EGFR and ERBB2 activity are often jointly active or inactive, consistent with the obligate dimer behavior of ERBB2. Ultimately, BCN1359 went on to become a pathologically complete responder. However, KSTAR predictions suggest that response may have been a product predominantly of the chemotherapy component of the treatment or that the pre- and post-treatment samples were possibly swapped at some point in the study.

## Discussion

We set out to develop a robust kinase activity inference method that is generally useful for all phosphoproteomic pipelines, especially clinical phosphoproteomics. The application of KSTAR to diverse experiments supports the idea that the KSTAR score is a reflection of kinase activity - the score increases with increasing kinase activities and decreases as a result of direct kinase inhibition. We found that these continuous-valued scores, up to the limit of statistical saturation, can identify the most active kinases within tissues and differences in kinase activities between tissues. These results also suggest that activity scores are comparable across kinases within an experiment, as the kinases with the highest activity scores in each tissue tended to be the ones you would expect, such as HCK and BTK in chronic myeloid leukemia cells and EGFR and ERBB2 in non-small cell lung carcinoma. This attribute is a result of creating uniform network sizes for all kinases (treating tyrosine and serine/threonine kinases independently)—if more phosphorylation sites were observed in one kinase's network, compared to another kinase network, then the statistical score reflects that as higher activity for the first kinase.

Among available kinase activity prediction algorithms, KSTAR is unique for several reasons. First, KSTAR is capable of utilizing any mass spectrometry experimental pipeline as evidence of kinase activity, irrespective of the type of quantification approach used, or in the absence of quantitative data. KEA3 is the only other algorithm that offers a similar flexibility, but we observed very poor performance of KEA3 in predicting physiologically relevant kinase ranks, such as an inability to produce results that improved tissue similarity/differences, and KEA3 results often suggested that the same kinase was high-ranking in both the most upregulated and downregulated of sites in an experiment. Second, KSTAR provides both a measure of degree of activity (score) and a binary cutoff indicating the significance of that score, allowing for a clear definition of an active kinase. KSEA and PTM-SEA also provide a binary call based on statistical significance, but these algorithms require differential quantification, perform significantly worse for tyrosine kinases, and are very sensitive to data losses within both tyrosine and serine/threonine networks (particularly losses of well-studied sites). Lastly, and perhaps most importantly, KSTAR accounts for the undue influence of well-studied sites at both the kinase and experiment level. We believe that existing activity inference approaches are triply hit by study bias issues: (1) they tend to rely on annotations, which are only available for well-studied kinases and phosphorylation sites, (2) phosphorylation sites identified in a phosphoproteomic experiment are more likely to be well-studied and most algorithms (except KARP) do not attempt to account for that, and (3) more well-studied sites tended to undergo larger fold-changes in the benchmarking data and therefore exhibit higher influence on predictions reliant on relative quantification (Supplementary Note 2). Hence, we believe that the algorithmic developments of KSTAR, such as enabling larger inclusion of phosphorylation evidence from experiments, kinase- and experiment-specific false positive rate control, and lack of dependence on fold-change or quantitative data, has

created an algorithm that is more broadly useful across kinases available in NetworKIN, but especially tyrosine kinase networks, which generally demonstrate more issues with study bias[25].

Finally, in our motivation to use kinase activity inference in a clinical setting, we demonstrated that KSTAR activity inference of tumor phosphoproteomic profiling can complement current clinical markers for HER2-status. KSTAR accurately identified a false HER2-positive patient and two HER2-pseudopositive patients in the microbiopsy study from Satpathy et al.[4]. KSTAR also correctly predicted that a clinically diagnosed HER2-negative PDX model would be responsive to EGFR/HER2 therapy, suggesting that phosphoproteomics combined with KSTAR can offer an orthogonal clinical diagnostic for breast cancer patients. On average, across all breast cancer samples analyzed by KSTAR in this study, roughly 30% of HER2-negative patients might have basal levels of HER2-activity, suggesting the possibility of a therapeutic avenue not normally offered HER2-negative patients.

## Methods

### Data preparation and mapping

All phoshoproteomic datasets used in this study were downloaded from the relevant journal site (typically found as supplementary data or tables in Excel). We identified the datasheet columns (a protein accession and a peptide indicating the phosphorylation site or an amino acid position) that allowed us to map the specific protein and phosphorylation site to the central ontology developed in KinPred[25]. If Uniprot accessions were not available, we used Uniprot web services to map to the Uniprot accession. KSTAR has a mapping function that matches the protein record and site information to the central KinPred ontology and appends two confirmed columns in that same ontology. Failure to map the protein or the site results in exclusion of that peptide from the mapped experiment, which happens in about 1 to 10% of peptides in any given dataset, an event that is logged. Also logged by mapping is if the site position is altered from the dataset to the ontology in cases where the reference protein is different than the database representation used by the authors of the study. The mapped files and their error and warning logs are provided in Supplementary Data available on FigShare (10.6084/m9.figshare.14919726).

In the experiments where relative quantification of phosphosites across multiple conditions was available, we used a relevant threshold to include or exclude specific sites as evidence in that particular experimental condition, while still ensuring that every experiment contained a sufficient number of sites to be statistically powerful (generally 50 or more sites for tyrosine, 1000 or more sites for serine and threonine). These thresholds, their meaning, and the resulting evidence sizes for experiments for all datasets are given in Supplementary Table 2 and these experiments are stored in KSTAR outputs as binary evidence used to generate kinase activity predictions (also available in FigShare KSTAR Supplementary Data).

### KSTAR algorithm implementation and details

All parts of the KSTAR algorithm, including pruning, mapping, activity inference, and plotting were implemented using Python3. The KSTAR code is highly customizable regarding the networks used, the limits on when to stop network pruning, the number of random experiments, and the number of decoy runs to use in FPR calculation. However, here we used the same parameters for all analyses as described and all data and code are provided on open source platforms (GitHub and Figshare), described below.

The 50 heuristically pruned networks were generated according to the rules discussed in Algorithm Overview, where tyrosine kinases had a finishing limit of 2000 substrates per kinase and serine/threonine kinases had a finishing limit of 2500 substrates per kinase. Further, substrates could be connected to no more than 10 tyrosine kinases or 20 serine/threonine kinases. The final networks also guaranteed that each kinase had the same proportion of substrates based

on "study bias" according to the total number of compendia substrates are annotated by and as defined by KinPred[25]. Substrates can be found in none or any of the five compendia included in ProteomeScout[51]: UniProt[52], phosphoELM[53], HPRD[54], dbPTM[55], or PhosphoSitePlus[56]. The networks used were downloaded from KinPred version 1.0 and based on the February 2020 reference phosphoproteome. The final networks used in this work are available on Figshare at https://figshare.com/account/projects/117123/articles/14944305.

We generated 150 unique random experiments for every experiment under consideration and such that the distribution of "study bias class" was the same as the real experiment, where study bias was defined as having low, medium, or high study bias based on being annotated in 0, 1-2, or 3-5 compendia. Unlike network generation, which uses the exact number of compendia annotations in the definition of study bias, random sampling groups the categories into classes, in order to guarantee true random sampling from the high study bias categories, since the number of total substrates annotated by 5 compendia is small. The false positive rate of the p-value observed in an experiment for a kinase is then calculated as the proportion of random experiments that had that p-value or more significant across the set.

Statistical tests, including the hypergeometric test and the one-sided Mann-Whitney U test, were performed using functions found within the stats module of the SciPy Python package[57]. For cases where datasets were large, such as for the two global phosphoproteomics studies[35,37] or for large serine/threonine networks, the KSTAR activity predictions were performed using a highly parallel version of the original code, implemented using the Nextflow software package[58].

### Compiling datasets for benchmarking analysis

In order to benchmark as many kinases and kinase families as possible, we sought to compile datasets across a wide range of studies and stimulation/inhibition conditions. Our final benchmarking dataset shares overlap to the one compiled by Hernandez-Armenta et al.[26], but has a larger breadth of tyrosine kinases profiled, as well as a few different serine/threonine kinases. For tyrosine kinases, 20 conditions were tested impacting a total of 19 tyrosine kinases, compiled across 8 studies[29,30,34,59–62]. For serine/threonine kinases, 31 conditions were tested impacting a total of 38 kinases, compiled across 10 studies[35,62–70]. All data were used as provided by the original authors, with the fold-change relative to the control used in the study, typically untreated cells. When assigning the expected perturbed kinases, we focused on kinases that were either directly targeted (such as EGFR during EGF stimulation) or had well characterized downstream activity (such as ERK during EGF stimulation). In studies where chemical proteomics data was available[35,60], we defined the expected perturbed kinases as the top 5 kinases based on strength of interaction from the chemical proteomics data (unless the particular drug interacted with fewer than 5 kinases). This allowed us to expand the benchmarking dataset to kinases that are often difficult to directly target and measure. For specific details about each dataset used in this work, see Supplementary Table 3 and Supplementary Table 4.

### Applying available kinase activity inference algorithms

**KSEA.** KSEA was implemented in Python3 as described in the original publication[39] and at (https://github.com/casecpb/KSEAapp/) according to using the z-score transformation to calculate the enrichment of substrates for each kinase:

$$\text{score}_k = \frac{(\bar{s} - \bar{p})\sqrt{m}}{\delta} \tag{1}$$

where $\bar{s}$ is the mean log2fold abundance of the substrates of kinase k, $\bar{p}$ is the mean log2fold abundance in the entire dataset, m is the number of substrates of kinase k identified in the dataset, and $\delta$ is the standard deviation of log2fold abundance across the dataset.

Significance is assessed using a right-tailed test and Benjamin-Hochberg FDR correction is applied to obtain the final list of significant kinases. Kinase-substrate annotations used for prediction were downloaded from PhosphoSitePlus[56], and mapped to KinPred as described in previous section. We compared our implementation of KSEA results to the available KSEA webapp[71] and found them to be consistent overall, but our mapping to PhosphoSitePlus (November 2021) increased the number of annotations in general, compared to the webapp (2016 PhosphoSitePlus data).

**PTM-SEA.** PTM-SEA was implemented within Rstudio using the R-gui provided in the ssGSEA2.0 github repo (https://github.com/broadinstitute/ssGSEA2.0). Annotations were downloaded from PTMsigDB v1.9.0[19]. All default parameters were used, except that the minimum number of substrates required (called min.overlap in the gui) was reduced from 10 to 1 to expand the number of total available predictions. To assess the impact of this substrate requirement, we also obtained which kinases maintained predictions when requiring at least 10 substrates to be present and plotted the results in Supplementary Note 4.

**KARP.** KARP was implemented in Python3 as described in Wilkes et al.[22], where the K-score for a particular kinase was generated using the below equation:

$$K = \frac{\sum_{i=1}^{m} \alpha_i}{\sum_{j=1}^{n} \beta_j} * \left(\frac{m}{t}\right)^{1/2} * 10^6 \tag{2}$$

where $\alpha$ and $\beta$ are normalized intensities for an individual phosphorylation site, $n$ is the total number of phosphorylation sites in the dataset, $m$ is the number of substrates identified in the dataset associated with the kinase, and $t$ is the total number of phosphorylation sites associated with the kinase in PhosphoSitePlus. As was done for KSEA, kinase-substrate annotations were downloaded from PhosphoSitePlus[56].

In order to identify the most perturbed kinases for a particular condition in the benchmarking dataset, the difference between the K-score for the experimental condition and the control was calculated, and kinases were ranked based on this difference. A positive difference indicated an increase in activity, while a negative difference indicated a decrease in activity. For applying KARP to the CPTAC data in Supplementary Note 6, the provided relative intensities were used to generate K-scores for each patient sample.

**KEA3.** In KEA3, kinases enrichment rankings are obtained for each of 11 different protein-protein interaction (PPI) or kinase-substrate interaction (KSI) databases based on the results of a Fisher's exact test. The mean rank is then obtained by averaging the ranks from each database, and it is the mean ranks that produce the final kinase enrichment ranking.

To obtain results from KEA3, gene names were obtained for each protein identified in a sample either from the original dataset itself or by using the UniProt web services to convert Uniprot accessions to the correct gene name. These gene names were then run through either the KEA3 web app or the KEA3 API via python3. To make results directly comparable to KSTAR results in figures 3–5, the same sites/genes used for KSTAR predictions were used for KEA3. Further, for the results in Fig. 5d, kinases without predictions in KSTAR were removed and kinases were then ranked across these 50 kinases based on their mean rank. Notably, the removed kinases include many serine/threonine kinases, which are not relevant to the phosphotyrosine enriched datasets where KEA3 is applied, hence the removal of non-overlapping kinases in KEA3 ultimately resulted in more relevant predictions that were dataset specific.

## Calculating $P_{hit}$

In this work, we have defined accuracy in a similar manner as in Yilmaz et al.[40], where the accuracy, $P_{hit}$, is defined as the fraction of times in which an expected positive kinase is found to be differentially active. We utilized two seperate indicators of a kinase hit: (1) kinase rank, where the perturbed kinase is found in the top 10 most differentially active kinases, or (2) statistical significance of activity, where the associated activity score was identified as statistically significant ($p \le 0.05$). A miss is then any instance for which a kinase is expected to be perturbed but is not defined as a hit.

$$P_{hit} = \frac{hits}{hits + misses} \tag{3}$$

We made an additional modification to the approach from Yilmaz et al., where we measured accuracy as a function of individual kinase performance, instead of as a function of individual experiment performance (Fig. 3b, Supplementary Note 4). We made this change due to the fact that both our benchmarking dataset and those compiled by others tended to be over-represented by certain kinases, typically ones that are easily targetable and/or well studied[26]. For example, in our dataset, AKT1 is perturbed across 13 different experiments while ATM is only perturbed in 2 experiments, contributing to 12% and 1.6% of tests, respectively (Supplementary Note 4). Given that this is the case, an algorithm that effectively predicts AKT1 activity, but not ATM activity, will appear to be successful due to the high influence of AKT1. Instead, by calculating the kinase-specific accuracy (fraction of expected perturbations of an individual kinase that were defined as a hit) and then averaging these scores to obtain the global accuracy in Fig. 3a, we can then obtain a global accuracy assessment that is more reflective of performance for all kinases present in the benchmarking dataset. In practice, this looks like the below two equations:

$$P_{hit,k} = \frac{hits_k}{hits_k + misses_k} \tag{4}$$

$$P_{hit,global} = \frac{P_{hit,k_1} + P_{hit,k_2} + \dots + P_{hit,k_n}}{n} \tag{5}$$

where $P_{hit,k}$ is the average accuracy for kinase $k$ across conditions where it is expected to be perturbed and $n$ is equal to the total number of kinases profiled in the benchmarking dataset (split by tyrosine and serine/threonine kinases).

## Random and targeted data loss experiments

To generate the random and targeted attack loss curves shown in Fig. 4b and Supplementary Note 4, we first identified the conditions in the benchmarking data where a statistically-based algorithm (KSTAR, KSEA, andPTM-SEA) found the perturbed kinase to have statistically significant differential activity (hits in Fig. 3) in the dataset. For each of these conditions, we generated a set of reduced experiments ranging from 5% to 95% data loss at increments of 5% data loss (5%, 10%, 15%,....). We repeated this process five times to obtain multiple replicates. KSTAR, KSEA, and/or PTM-SEA (depending on if initial activity was significant) were then applied to all of the reduced experiments to regenerate predictions at each level of data loss, and the average false discovery rate curve was obtained by averaging across the five replicates.

Datasets were reduced in two ways. For the random attack/removal, sites were removed at random so that all sites were equally likely to be removed from the experiment with each replicate. For the semi-random targeted attack sites were first organized based on the number of compendia they are recorded (an indicator of study bias). The most well-studied sites, which are found in all five compendia discussed in this work, are chosen at random for removal. If all of these

sites are removed, the next most well-studied sites, which are found in four of the five compendia, are chosen at random for removal. This process continues until the correct amount of sites has been removed from the experiment. As the targeted attack is only a semi-random process, variance across the replicates tended to be smaller than for the random attack.

Once all random and targeted attack curves were generated, tolerable loss and sensitivity could be calculated as described in Fig. 4a. Tolerable loss was obtained using the entire loss curves (0 to 95%), while sensitivity was calculated using only half of these curves (0 to 50%).

### Comparing phosphoproteomics datasets

When comparing different phosphoproteomics datasets, all data was first mapped to KinPred, as described above. After mapping, the set of phosphosites identified in each dataset was obtained, where any site present in a sample, regardless of relative quantification, was considered in the set. The similarity of two sets of sites were compared using the Jaccard index, defined by dividing the intersection of the two sets (number of sites identified in both experiments) by the union (the total number of unique sites identified across both experiments). To compare these same experiments based on their KSTAR profiles, we used Spearman rank correlation, as implemented in the SciPy Python package[57].

### Reporting summary

Further information on research design is available in the Nature Research Reporting Summary linked to this article.

## Data availability

The networks generated in this study have been deposited as data files in Figshare at 10.6084/m9.figshare.14944305. The Resource Files, along with network files, required to reproduce the experiments in this work have been deposited in Figshare at 10.6084/m9.figshare.14885121.v6. All source data have been deposited within the KSTAR Figshare project https://figshare.com/projects/KSTAR/117123 at the permanent doi 10.6084/m9.figshare.14919726. We have provided ancestry, age, and sex for cell lines used as models in published experimental studies analyzed in this work in Supplementary Table 7. For patient-focused demographic information, please reference the original publications (Fig. 6a: Supplementary Table 1 in Mertins et al.[5], Fig. 6b: Supplementary Data 1 in Huang et al.[8], Fig. 6c: Supplementary Table 1b from Satpathy et al.[4]). Source data are also provided as a Source Data file. Source data are provided with this paper.

## Code availability

All code for KSTAR are available at our Git repository: https://github.com/NaegleLab/KSTAR with a snapshot of the repository at the time of this work at 10.6084/m9.figshare.20146016. Code for applying KSTAR to data, including all examples within this paper, is available at our KSTAR Applications Git repository: https://github.com/NaegleLab/KSTAR_Applications with a snapshot of the repository at the time of this work at 10.6084/m9.figshare.20146013.

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

## Acknowledgements

Research reported in this publication was supported by the National Cancer Institute of the National Institutes of Health under Award Number R21CA231853 (K.M.N). The content is solely the responsibility of the authors and does not necessarily represent the official views of the National Institutes of Health. The authors acknowledge Research Computing at The University of Virginia for providing computational resources and technical support that have contributed to the results reported in this publication (https://rc.virginia.edu).

## Author contributions

S.C., B.J., and K.M.N. conceived of and implemented the algorithm and analyses. H.A. contributed to analyses of other algorithms. C.X.M. contributed to analysis and insight in breast cancer applications. S.C. and K.M.N. wrote the manuscript.

## Competing interests

The authors declare no competing interests.
