## [Peer Review File · Nature Communications]

REVIEWER COMMENTS

Reviewer #1 (Remarks to the Author):

In this manuscript, the authors developed network-based algorithm named KSTAR to count the number of phosphorylation sites of a given kinase against random permutations from the phosphoproteomic data, as a computational measure of “net kinase activity”. They tested their method on the phosphoproteomic data sets of various cell lines, including mammary epithelial 184A1 cells, Jurkat cells, K562 cells, breast cancer BT-474 cells, HCT116 cells, as well as the tumor samples of 107 breast cancer patients from CPTAC, and the data of PDX models of breast cancer. Indeed, KSTAR can correctly capture some really functional kinases from each of the data set. However, the fundamental flaw of the study is that the basic hypothesis is not novel, and the method is technically problematic. The authors do not compare KSTAR to other existing methods in a justified manner, and new biological findings are not reported. Due to lack of novelty, this study has no possibility to be considered by NC. My major concerns are shown as below:

1. The authors are not experts in the field of post-translational modification (PTM) Bioinformatics, or computational kinase biology. Thus, I think that they did not notice that their basic hypothesis, “...the more active a kinase is, the more of its substrates will be observed in a phosphoproteomic experiment...”, was not novel at all. In 2012, Song et al. conducted a Yates’ chi-squared test to identify kinases with more or less phosphorylation sites by comparing the phosphoproteomic data of a pair of samples (Mol Cell Proteomics, 2012, 11, 1070-83. PMID: 22798277). Later, this test was implemented into an applicable method named kinase activity analysis (KAA), and its hypothesis is “...a kinase with higher activity will phosphorylate more sites” (Mol Cell Proteomics, 2014, 13, 3626-38. PMID: 25293948). This hypothesis was also adopted in development of kinase-substrate enrichment analysis (KSEA) for analysis of quantitative phosphoproteomic data, and the hypothesis was described as “...the amounts of several substrates of the same kinase (thus diluting the contribution of outliers) could be used to gain reliable information on the activity of a kinase pathway from MS-based phosphoproteomics” (Sci Signal, 2013, 6, rs6. PMID: 23532336). Again, this hypothesis was also adopted for development of kinase enrichment analysis (KEA), and the hypothesis was re-stated as “...The enrichment of known kinase substrates in a set of differentially phosphorylated proteins can serve as a potential marker of the upstream kinases' state and provide insights into physiological and pathophysiological mechanisms” (Bioinformatics, 2009, 25, 684-6. PMID: 19176546; Nucleic Acids Res, 2021, 49, W304-W316. PMID: 34019655).

2. The implementation of KSTAR is quite similar to KEA3 (<https://maayanlab.cloud/kea3/>), which was developed in an easy-to-use manner. A heuristic prune as well as a random permutation did not significantly increase the originality of the study. Indeed, KEA/KEA3 is useful to find functional kinases, it can be expected that a slightly modified form, KSTAR, can also predict some ones.

3. The fundamental flaw of KAA, KSEA and KEA is that the quantitative values of the phosphoproteomic data were not considered. There are a lot of studies that have been focused on using both the numbers and quantitative values of phosphorylation sites. For example, a z-test-based method also named KSEA could be used for such analyses (Bioinformatics, 2017, 33, 1845-1851. PMID: 28200105).

4. Another important problem of KSTAR is that the phosphoproteomic data of the control was not considered. I do not think a random permutation can faithfully reflect the bona fide phosphorylation state in control samples. To predict patient-specific kinase activities, using paired non-cancerous adjacent tissues (NATs) can efficiently remove the genetic heterogeneity of different patients. Except the authors, all scientists in the field of PTM Bioinformatics will never neglect the control data.

5. If the authors can incorporate the quantitative values, and the control data in their KSTAR, I think their study might have a chance to reach the Bioinformatics or MCP level, because the novelty is not very high. I understand that perhaps the authors are really not familiar with the current progresses in this field, but they should conduct a careful literature reading before starting the project.

6. The authors stated that “thresholded networks” were used in previous studies. However, their heuristic pruning generated thresholded networks. They changed nothing.

7. The authors did not compare their method to other existing methods in a justified manner. I do not think a slightly modified version of KEA can generate some really important biological findings.

8. In KEA3, the authors used “kinase state” rather than “kinase activity”, because kinases can physically interact and phosphorylate their substrates. The change of numbers of phosphorylation sites might be either attributed to the change of kinase activity, or the kinase interacting ability.

Reviewer #2 (Remarks to the Author):

The authors present a new method for kinase activity estimation. The main novelty lies in the fact that rather than using known kinase-substrate interactions, they rely on predicted networks from the algorithm NetworKIN. For this they have developed a statistical framework to make use of these

predictions and extract signal from the noise. They then have applied the method to different applications with an especially interesting one on identifying patients that displayed kinase activities different from their HER2 status and thus would need to be treated differently than what the standard of care suggests.

The method is an addition to several other methods predicting kinase activities (here the literature review could be more comprehensive, only the Cutillas group and INKA method are mentioned), but has the benefit that doesn't need 'differential' abundance of peptides for input and so has the long-term potential to be more applicable e.g. in clinical samples where a paired normal is missing etc. The use of the predicted network as a basis for the kinase activity predictions also somewhat reduces the literature bias that comes from the annotation of only well studied kinase-substrate pairs and thus presumably allows predictions to be made for more datasets than is possible with previous methods.

Overall, the method is interesting, and I especially like some of the applications where they demonstrate the usage.

Major concerns

1. The main concern I have is that there is nowhere a comparison with the other methods for kinase activity prediction. There should be a systematic comparison across all the applications presented demonstrating how KSTAR performs differently (and better?) from the other existing methods, and thus showing the benefit of using the predicted network as a basis vs the known substrates. In particular, in all examples presented extremely well-studied kinases are studied/highlighted, which already have several known substrates and I suspect (though would like to be proven wrong) that the results of KSTAR would be very similar to the other algorithms doing the same job. And I would expect the same also from the cross-dataset comparison that the authors present.

2. Secondly, other than anecdotal examples (even if they are nice ones) there is no systematic evaluation of the method. I understand that this can be tricky, but for example for the kinase inhibition section there are at least two datasets from the Cutillas group (Wilkes et al and Hijazi et al) that could have been used, with a large number of kinase inhibitions (22 and 103 respectively) to evaluate the performance and the ability of KSTAR to recover the inhibited kinase as the top downregulated one. This would be a very straightforward evaluation that would include many samples and could provide a quantitative performance metric. They could also (optionally) present the profiles of kinase inhibition across the different cell lines where the kinase inhibitors are the same to show the different responses of different cell lines to these inhibitors.

3. For the Jurkat example it seems suspicious to me that the FYN, LCK, ITK, BTK, HCK kinases seem to have the same activity across a time span of 60 mins. I am pretty sure they are more dynamic going up in the first 5 mins and then definitely winding down at later time points, though I can't recall now exactly when that would be. Either way it is odd signalling-wise to have these tyrosine kinases on for 60 minutes at the same level. Perhaps the authors can double check this result and see if there is any literature supporting it, or mention it in the discussion. Similarly, it is not clear to me if the KSTAR score is comparable across the kinases or not. i.e. is an EGFR score of 10 less than an HCK score of 15 or can we not compare across datasets (this is just for my curiosity).

Minor comments:

1. The supplementary figures are a mess, with multiple figures under each figure and no legend making it difficult to follow what is going on where and what the main message is, as well as how it corresponds to the text.
2. Using the NetworKIN predicted network as a background doesn't really mitigate the literature biases that much since NetworKIN relies on the STRING network and position specific scoring matrices for their predictions both of which suffer from bias. Perhaps this can be mentioned in the discussion since they authors stress that their method is bias-aware as a feature.
3. The heuristic algorithm for pruning the networks could be described in a clearer way

Reviewer #3 (Remarks to the Author):

Kinases play an important role in oncogenesis and cancer progression and are therefore often used as drug targets for cancer therapies. However, treatment challenges remain, such as the development of resistance and non-response to treatment, which could be reduced by reliable estimation of kinase activities.

In this manuscript the authors present KSTAR, an algorithm that predicts kinase activity profiles from phosphoproteomic data using kinase-substrate prediction networks. It is based on the assumption that the more active a kinase is, the more of its substrates will be observed in a phosphoproteomic experiment and additionally tries to handle the kinase-substrate connections in an error- and bias-aware manner. For that, kinase-substrate prediction graphs from NetworKIN are pruned instead of using thresholded networks to overcome high study bias, isolate less phosphorylation sites and remove high kinase network overlap. The authors validate KSTAR by predicting kinase activities from different perturbation experiments and testing the robustness of the prediction by using data across

different studies and laboratories. Additionally, KSTAR was used to identify clinically-relevant kinase activity profiles using bulk cancer biopsy data. According to this validation, KSTAR was able to reproduce expected kinase activity profiles, show robust predictions across studies from different laboratories allowing a better comparison, identify tissue-specific kinase profiles and could complement current clinical standards in the example of HER2-status identification.

The main advantage of KSTAR is that by only relying on the identification of phosphorylation sites from a sample instead of requiring relative quantification of phosphorylation sites it is able to include more phosphosites and no matched samples are required. Furthermore, it addresses study biases in the kinase-substrate network.

This could be a meaningful contribution to the field, once the following comments are adequately addressed:

Major points

1. Our main criticism is that this method is never compared to other current approaches that take the quantification of phosphorylation sites into account or use thresholded networks. It would be essential to see how well this method performs compared to others to be able to truly assess its value.
2. We think it could be useful to include further benchmarking datasets for the validation of KSTAR as in e.g. Kuleshov et al, *Nucleic Acids Res.*, 2021, Yilmaz et al, *Nature Communications*, 2021, Hernandez-Armenta et al, *Bioinformatics*, 2017, or others.
3. The authors used the kinase-substrate graph of NeworKIN. However, this resource is relatively old so maybe it would be good to test the prediction using more recent resources.

Minor points

1. In the method section "KSTAR algorithm implementation and details" the last sentences of the first two paragraphs are cut off (line 466, line 473).
2. Authors state: "NeworKIN results in predictions consistent with physiological expectation, but using kinase-substrate networks from other prediction algorithms (GPS and PhosphoPICK), did not result in physiologically relevant hypotheses" - could authors elaborate on this?

We would like to thank the reviewers and the editor for their time and insights. We believe that the response has greatly strengthened the paper, especially for a broader audience, and increases context for where KSTAR fits in the field of kinase activity prediction.

Overview of major changes:

1. Introduction - now includes explicit connection to existing algorithms and how KSTAR is different. It also includes our additional reasoning for not relying on quantification from mass spectrometry pipelines (in addition to arguing for clinical relevance).
2. Increased clarity and supplements on algorithm for heuristic pruning and the results that demonstrate the algorithm's characteristics for creating desired characteristics.
3. Improved supplemental layouts.
4. Additional results and supplements directly comparing performance in two popular algorithms (KSEA and KEA3)
5. Revised Discussion - focusing less on reiteration of results and more on what makes KSTAR a major and novel contribution to the field (interpretable, quantitative, physiologically relevant, clinically relevant, and usable on any type of dataset) and highlighting the problems with benchmarking approaches and the presence of study bias in other algorithms.

Reviewer #1 (Remarks to the Author):

In this manuscript, the authors developed network-based algorithm named KSTAR to count the number of phosphorylation sites of a given kinase against random permutations from the phosphoproteomic data, as a computational measure of "net kinase activity". They tested their method on the phosphoproteomic data sets of various cell lines, including mammary epithelial 184A1 cells, Jurkat cells, K562 cells, breast cancer BT-474 cells, HCT116 cells, as well as the tumor samples of 107 breast cancer patients from CPTAC, and the data of PDX models of breast cancer. Indeed, KSTAR can correctly capture some really functional kinases from each of the data set. However, the fundamental flaw of the study is that the basic hypothesis is not novel, and the method is technically problematic. The authors do not compare KSTAR to other existing methods in a justified manner, and new biological findings are not reported. Due to lack of novelty, this study has no possibility to be considered by NC.

My major concerns are shown as below:

1. The authors are not experts in the field of post-translational modification (PTM) Bioinformatics, or computational kinase biology. Thus, I think that they did not notice that their basic hypothesis, "...the more active a kinase is, the more of its substrates will be observed in a phosphoproteomic experiment...", was not novel at all. In 2012, Song et al. conducted a Yates' chi-squared test to identify kinases with more or less phosphorylation sites by comparing the phosphoproteomic data of a pair of samples (Mol Cell Proteomics, 2012, 11, 1070-83. PMID: 22798277). Later, this test was implemented into an applicable method named kinase activity analysis (KAA), and its hypothesis is "...a kinase with higher activity will phosphorylate more sites" (Mol Cell Proteomics, 2014, 13, 3626-38. PMID: 25293948). This hypothesis was also adopted in development of kinase-substrate enrichment analysis (KSEA) for analysis of quantitative phosphoproteomic data, and the hypothesis was described as "...the amounts of several substrates of the same kinase (thus diluting the contribution of outliers) could be used to gain reliable information on the activity of a kinase pathway from MS-based phosphoproteomics" (Sci Signal, 2013, 6, rs6. PMID: 23532336). Again, this hypothesis was also adopted for development of kinase enrichment analysis (KEA), and the hypothesis was re-stated as "...The enrichment of known kinase substrates in a set of differentially phosphorylated proteins can serve as a potential marker of the upstream kinases' state and provide insights into physiological and pathophysiological mechanisms"

(Bioinformatics, 2009, 25, 684-6. PMID: 19176546; Nucleic Acids Res, 2021, 49, W304-W316. PMID: 34019655).

We stated our algorithm was novel, not the connection of the hypothesis to the function of kinases. Since kinases fundamentally only work in one way, catalyzing the transfer of phosphate at a rate to substrates that is proportional to its activity and its specificity, then it really is only feasible to connect the mathematical approach to its physiological function (either more substrates, more of the same substrate, or some combination of both). We believe that our increased comparison of our algorithm to others will create a deeper delineation for readers as to our particular approach and its novelty. Although, it seems also this reviewer is suggesting that some novel aspect of kinases would have to be discovered for anyone to publish in this field, a sentiment that is deeply troubling. The lack of methods that have successfully translated to the clinic suggests to us that there is a significant room for improvement in the field.

2. The implementation of KSTAR is quite similar to KEA3 (<https://maayanlab.cloud/kea3/>), which was developed in an easy-to-use manner. A heuristic prune as well as a random permutation did not significantly increase the originality of the study. Indeed, KEA/KEA3 is useful to find functional kinases, it can be expected that a slightly modified form, KSTAR, can also predict some ones.

We have included KEA3 as a direct demonstration, although the only similarity between KSTAR and KEA3 in the field is that KEA3 does not technically require differential or quantitative data (although, as stated by the authors, it is assumed the gene list provided is motivated from differential evidence). However, the reviewer is incorrect that KSTAR is similar to KEA3. KEA3 is a ranked list across an aggregated set of protein-protein interaction databases that completely disregards the actual phosphorylation evidence from an experiment. The separation of KEA3 from physiological relevance of a phosphoproteomic experiment is likely the reason that KEA3 predictions are not usable for gaining clinically or scientifically useful predictions based on the analyses in this work.

3. The fundamental flaw of KAA, KSEA and KEA is that the quantitative values of the phosphoproteomic data were not considered. There are a lot of studies that have been focused on using both the numbers and quantitative values of phosphorylation sites. For example, a z-test-based method also named KSEA could be used for such analyses (Bioinformatics, 2017, 33, 1845-1851. PMID: 28200105).

The reviewer is contradicting their self in this review - stating that KSEA both does not and does use quantitative information (it in fact does and relies on it), both the original GSEA-based approach and the z-test based update. Also, we should note that the reviewer made no actual connection between their statements and the work under review in this statement. However, we assume the reviewer is arguing that algorithms that use quantification are superior and we disagree with that. In addition to multi-sample quantification not being relevant to clinical use, we believe that such algorithms are more likely to be subject to study bias issues, which we have noted in the discussion. Most importantly, we have added in the introduction the severe limitation that quantification (of any type that does not include an absolute spike-in) cannot be used for comparisons between peptide abundances, which these algorithms mathematically rely on the assumption that they do, a fundamental flaw for algorithms relying on relative quantification. Here is the additional text we added with respect to that:

“However, generating kinase activities from this data requires overcoming several challenges, including: 1) data sparsity or missing data -- in shotgun phosphoproteomics (i.e. discovery-based approaches) lack of detection of a phosphorylation site may not be evidence that it is not present in the sample and although CPTAC approaches often using a reference standard, only 5 to 15% of all phosphorylation sites are common across patient cohorts, 2) there is an extreme paucity of data regarding the direct connection between phosphorylation sites and their kinases (only 5% of

phosphorylation sites are annotated with a kinase) \cite{Needham2019}, and 3) challenges in relating quantitative data available from phosphoproteomics experiments to kinase catalytic activities. Quantification is particularly challenging -- unless one uses a known spike-in for absolute quantification for every phosphopeptide of issue, all quantification (label and label-free) is relative -- i.e. it is not possible to understand differences in quantities between peptides, only differences of a peptide between conditions. This is due in large part to peptide-specific sample losses and ionization \cite{Solari2015a}. For example, if there are two kinases, each with a different substrate and one changes from 1fmol to 2fmol, but the second substrate changes from 8pmol to 16pmol, there is a 4,000-fold difference in catalytic activity between the two kinases, but relative quantification interprets both of those as the same (2-fold different)."

4. Another important problem of KSTAR is that the phosphoproteomic data of the control was not considered. I do not think a random permutation can faithfully reflect the bona fide phosphorylation state in control samples. To predict patient-specific kinase activities, using paired non-cancerous adjacent tissues (NATs) can efficiently remove the genetic heterogeneity of different patients. Except the authors, all scientists in the field of PTM Bioinformatics will never neglect the control data.

We had a difficult time identifying the scientific merit in this comment. The random permutation we performed was to control entirely for the false positive rates of the underlying study bias. We in no way, shape, or form propose these are control datasets representing a patient's healthy tissue. Also, we never propose anywhere that we seek to identify dysregulated kinases explicitly from expected behavior, ergo a healthy matched control is not relevant. Finally, the idea of having a matched control runs counter to creating something that might eventually be diagnostically useful, since matched controls are rare in the clinic.

5. If the authors can incorporate the quantitative values, and the control data in their KSTAR, I think their study might have a chance to reach the Bioinformatics or MCP level, because the novelty is not very high. I understand that perhaps the authors are really not familiar with the current progresses in this field, but they should conduct a careful literature reading before starting the project.

We deeply disagree with this reviewer regarding their assessment of "novelty". Likely, the incorrect assessment of this reviewer is related to our initial lack of comparison to other approaches in the field and we hope our increased attention to this detail in the introduction, results, and discussion, will improve the reviewer's understanding of this work and others.

6. The authors stated that "thresholded networks" were used in previous studies. However, their heuristic pruning generated thresholded networks. They changed nothing.

We hope the increased clarity introducing the algorithm and greatly expanded supplement highlighting the approach and pseudocode will help this reviewer and other readers understand the basic tenets of graph theory in order to understand why there is a fundamental difference between arriving at a sparse network from a dense network through the process of thresholding and doing so through our novel approach to arriving at ensembles of graphs that represent likely relationships between kinases and substrates, set by heuristic probability and coupled with rules to mitigate the massive problems inherent to these networks.

7. The authors did not compare their method to other existing methods in a justified manner. I do not think a slightly modified version of KEA can generate some really important biological findings.

The reviewer is incorrect in the comparison of KEA to KSTAR (other than KEA being the only other algorithm that is agnostic to quantification approaches), these algorithms are in no way comparable. Increased and direct comparison within the paper hopefully will clarify the distinct differences.

8. In KEA3, the authors used “kinase state” rather than “kinase activity”, because kinases can physically interact and phosphorylate their substrates. The change of numbers of phosphorylation sites might be either attributed to the change of kinase activity, or the kinase interacting ability.

We agree with the reviewer, KEA3 should not call their “score”, which is an aggregate ranking of gene list interactions, a kinase activity score since it does not reflect kinase activity. We also use quotes, since there is no single score, but rankings, which to us, were unclear how to interpret. Regarding the rest of this statement, we would like to remind the reviewer that physical contact between kinases and substrates is always a necessary requirement for catalytic transfer of phosphate from the donor ATP molecule to the substrate. However, kinase activity is typically tightly regulated, so interaction alone through tertiary interactions is rarely sufficient (another reason we believe KEA3 is not particularly physiologically reflective as it depends only on protein-protein interaction evidence).

Reviewer #2 (Remarks to the Author):

The authors present a new method for kinase activity estimation. The main novelty lies in the fact that rather than using known kinase-substrate interactions, they rely on predicted networks from the algorithm NetworkKIN. For this they have developed a statistical framework to make use of these predictions and extract signal from the noise. They then have applied the method to different applications with an especially interesting one on identifying patients that displayed kinase activities different from their HER2 status and thus would need to be treated differently than what the standard of care suggests.

The method is an addition to several other methods predicting kinase activities (here the literature review could be more comprehensive, only the Cutillas group and INKA method are mentioned), but has the benefit that doesn't need 'differential' abundance of peptides for input and so has the long-term potential to be more applicable e.g. in clinical samples where a paired normal is missing etc. The use of the predicted network as a basis for the kinase activity predictions also somewhat reduces the literature bias that comes from the annotation of only well studied kinase-substrate pairs and thus presumably allows predictions to be made for more datasets than is possible with previous methods.

Overall, the method is interesting, and I especially like some of the applications where they demonstrate the usage.

Major concerns

1. The main concern I have is that there is nowhere a comparison with the other methods for kinase activity prediction. There should be a systematic comparison across all the applications presented demonstrating how KSTAR performs differently (and better?) from the other existing methods, and thus showing the benefit of using the predicted network as a basis vs the known substrates. In particular, in all examples presented extremely well-studied kinases are studied/highlighted, which already have several known substrates and I suspect (though would like to be proven wrong) that the results of KSTAR would be very similar to the other algorithms doing the same job. And I would expect the same also from the cross-dataset comparison that the authors present.

Comparison to Other Approaches

We greatly appreciate the reviewer suggestions for highlighting the differences between KSTAR and existing resources. In our original submission, we struggled with this and how to do it well, especially since our previous attempts with these algorithms were unsuccessful in most of the test cases we

approached (including those kinases that seem like they should perform well). We were unsure how to present this negative data well and so we left those findings out of our original submission. However, we appreciate the perspective of ensuring readers understand how this approach is different and what it can offer that others cannot and so our changes have focused on these specific details. Here are the changes we have made for comparison to other tools:

- **Major changes in the introduction** that frames how this algorithm compares. We included our reasoning (in addition to clinical relevance) for avoiding reliance on quantitative data.
- **Added a supplemental table** that details existing methods and summarizes accessibility, underlying networks, and general mathematical approach and data inputs.
- We **added direct comparison to two methods**, KSEA (z-score version) for datasets that include differential quantitative data (due to its popularity and ease-of-use), and KEA3 for all datasets (since it is the only algorithm, outside of KSTAR, that can be applied universally to any phosphoproteomic dataset, regardless of whether it used labeled, label-free, multi-sample, or single-sample approaches). These results have been discussed in the manuscript and added to the supplements.

For ease of review, here is a summary of these comparisons:

KSEA:

- KSEA performs very poorly in phosphotyrosine networks due to lack of data and dependence on annotations (also please note that the authors themselves already ran comprehensive analysis on the use of NetworkKIN thresholded networks, which performed worse, so we use annotations as they suggest).
- KSEA is not capable of predicting HER2 activity in any of the breast cancer samples we have due to too low of an annotation.
- It also performs poorly for some kinases in serine/threonine. For example, it cannot predict AKT2 or AKT3 activity in the AKT inhibition study due to low annotation coverage within the experiment.
- KSEA suffers from study bias as KSTAR did before we compensated for it. For example, KSEA systematically highlights non-physiological kinases as being upregulated, but corresponds with study bias (e.g. KSEA predicts LCK, a hematopoietic tyrosine kinase is the most upregulated in epithelial cells stimulated with epidermal growth factor).
- KSEA cannot be applied to half of the studies we used since they lack quantitative evidence.
- KSEA, both because of statistical power, but also reliance on fold-changes, cannot be used for predicting basal activity of HER2 as a complementary diagnostic to clinical FISH/IHC.

KEA3:

- Although KEA3 is the only other universally applied algorithm, and in that sense is the most comparable to KSTAR, it performed poorly across the board on these tasks and was difficult to interpret for use in anything scientifically or clinically relevant. Here are some of the details of that performance:
 - Systematic performance tests:
 - Figure 3: Although KEA3 rankings of kinases shows high similarity between independent datasets of similar tissues it also shows very high similarity between independent datasets of different tissues (this systematic comparison was included as a new panel Figure 3D). In contrast, KSTAR results maintain high similarity within tissue and very low similarity in kinase profiles between tissues.
 - KEA3 consistently highlights the same kinases as being both upregulated and downregulated. So, there is no consistency with the basic premise that if a kinase is highlighted in the selection of sites that come from a quantitative cutoff in an inclusion set, then it shouldn't show up in the exclusion set as well. However, KEA3 very regularly

suggests that the same kinase has a strong signature in both sets. In contrast, by its very statistical definition KSTAR is incapable of showing the same degree of activity in an inclusion and exclusion (i.e. when KSTAR activity goes down between basal and inhibition for AKT1, it's because it is "enriched" in the sites excluded).

- Although KEA3 gives some relevant results, especially since we limit KEA3 results to the relevant subtype of kinase for a given experiment, the type of information is difficult to interpret for useful scientific and clinical tasks. For example:
 - EGFR is the 10th-highest ranked kinase in response to EGF treatment, below a number of hematopoietic kinases (possibly also showing study bias issues in this approach).
 - KEA3 ranking of HER2 has no correlation with clinical HER2-status nor does it represent correlative changes in HER2 rank in response to HER2 therapy. We can simply find no way in which this approach could be clinically useful for diagnosing basal HER2 activity or failure to respond to therapy.

2. Secondly, other than anecdotal examples (even if they are nice ones) there is no systematic evaluation of the method. I understand that this can be tricky, but for example for the kinase inhibition section there are at least two datasets from the Cutillas group (Wilkes et al and Hijazi et al) that could have been used, with a large number of kinase inhibitions (22 and 103 respectively) to evaluate the performance and the ability of KSTAR to recover the inhibited kinase as the top downregulated one. This would be a very straightforward evaluation that would include many samples and could provide a quantitative performance metric. They could also (optionally) present the profiles of kinase inhibition across the different cell lines where the kinase inhibitors are the same to show the different responses of different cell lines to these inhibitors.

Systematic Comparison

We appreciate the perspective of both reviewers 2 and 3 that a systematic exploration to kinase inhibition, including the use of established datasets, would be, in theory, a way to measure the global performance of KSTAR and other predictors. We had considered this both before submission and in response to these comments. We agree that the simplistic viewpoint that a kinase inhibitor should result in large, or maximum, changes after inhibition to its targeted kinase would allow for the possibility for systematic evaluation of a prediction algorithm. However, for this to actually occur, the following constraints MUST be in place in each and every one of the collections of inhibition studies that were suggested by both reviewers (where there are fewer than it appears as multiple papers, e.g. Yilmaz et al, Nature Communications, also use Hernandez-Armenta et al. 2017): 1) the kinase must be expressed and must be active, 2) the inhibitor must be exquisitely specific and potent, downregulating that kinase by the largest amount (which circularly means it had to have a relatively high enough expression for changes to be noticeable). Additionally, the latter case assumes that: a) interconnected and downstream kinases are not impacted more than the target kinase and b) off-target effects are minimal. Yet, this is rarely the case, as demonstrated by our examples and available publications and the explicit development of therapies based on the idea of polypharmacology -- i.e. that therapeutic effectiveness of kinase inhibitors is through non-specific and non-obvious ways that rarely involve the precise and simplistic assumptions outlined above. Additionally, here is how each of our chosen examples directly counter the simplistic idea of a direct correlation between an inhibitor and the maximum downregulation of a targeted kinase:

1. Our BCR-ABL results of Fig. 2C are consistent with the publication and what is known about BCR-ABL inhibition – that ABL activity is so high, the therapeutic effectiveness in treating leukemia with dasatinib is not through substantial downregulation of ABL, but rather modest downregulation of ABL and massive downregulation of cross-talk from ABL to RTKs. Hence, a systematic evaluation as proposed by the field would suggest this result is a failure (ABL was not the most downregulated kinase), yet it is a success in identifying the ways in which dasatinib acts and can identify the physiological nuances of cancer therapy.

2. There are several examples given here where resistance (i.e. cell or patient context) where kinase activity does not fall (BRAF inhibition by the V600E-specific inhibitor and HER2-therapy response). These would be encoded as failures, but are useful, physiologically-relevant findings.
3. There are examples that demonstrate the assumption that a kinase starts out active in a study, and therefore can be seen as decreasing, are fraught. For example, we identified the three false positives in HER2-positive breast cancer. Yet, simplistic assumptions would count these examples as errors, since it would have been pre-determined by diagnosis that these are basally active HER2 tumors.

We also note that the **systematic evaluation datasets (including those from the Cutillas and Beltrao labs) are heavily skewed towards serine/threonine kinases and that despite a large number of datasets, the same kinases are often targets (we have included a new supplement to help visualize this problem)**. This suggests that using these datasets ultimately skews testing and performance towards very few kinases and we find that existing approaches are especially poor at handling and predicting tyrosine kinases, which is a category of kinases underrepresented in these large compendia datasets. Since tyrosine kinase inhibitors are a major class of FDA-approved drugs, it is important to do well in this class of predictions. We believe these systematic datasets have worked to obscure how poorly prediction algorithms are doing within tyrosine kinase activity prediction and we wish to avoid perpetuating this problem.

Therefore, in short, we propose that, given the complexity of kinases, their interconnectedness, the `dirtiness' of kinase inhibitors, the importance of tyrosine kinases, and the context-specific nature of kinase activity and therapy response that there is no substitution yet for the deep, detailed, and mechanistic evaluation of kinase prediction algorithms. **Since there is no way yet to truly ask "Are they right?", we instead have focused on "Is it useful?"**. We hope the reviewers concur that we would do the field a further disservice perpetuating simplistic views of biology, when biology is absolutely not willing to submit to simplicity and that these evaluations are not just "anecdotal", but absolutely scientifically necessary and one of our ways of biologically evaluating usefulness. In its stead, we have made the following **major changes to the Discussion**:

1. We have included an explicit discussion of this point and our choice to avoid oversimplifications through aggregated and large datasets for systematic testing. As in this letter, we include explicit examples that demonstrate the counterpoints to the idea that a simplistic encoding of accuracy can be assumed in such datasets and we have given readers a supplement that demonstrates the bias in these datasets for serine/threonine kinases and specific subsets of those kinases.
2. Instead, we introduce the idea of systematic evaluation by asking if transference from phosphoproteomic data, which generally is poorly suited to being comparable across samples, to kinase activity allows us to observe improvements in similarity and differences based on tissue types. **We thought this was a rather elegant way to approach systematic evaluation, which is scientifically accurate** – there is an expectation that kinase profiles of the same or similar cell lines and tissues should be more similar to each other than they are to distinctly different tissues – but, also makes no assumptions about what kinases should be highest or lowest or most different between tissues, but simply allows us to ask – Did the prediction of kinase activities stabilize sparse phosphoproteomic data in a useful way? In this regard, only KEA3 and KSTAR are even capable of such a test (at least in a tractable manner and given the data we had in hand) and KEA3 fails in this systematic analysis.

3. For the Jurkat example it seems suspicious to me that the FYN, LCK, ITK, BTK, HCK kinases seem to have the same activity across a time span of 60 mins. I am pretty sure they are more dynamic going up in the first 5 mins and then definitely winding down at later time points, though I can't recall now exactly when that would be. Either way it is odd signalling-wise to have these tyrosine kinases on for 60 minutes at the same level. Perhaps the authors can double check this result and see if there is any literature supporting it, or mention it in the discussion. Similarly, it is not clear to me if the KSTAR score is comparable across the kinases or not. i.e.

is an EGFR score of 10 less than an HCK score of 15 or can we not compare across datasets (this is just for my curiosity).

Jurkat: We agree, this would be surprising, but this is actually in **seconds**. So, we are seeing immediate and physiologically-relevant signaling within 5 to 60-seconds post-stimulation. We added extra text in the legend to note early and refers to seconds.

Comparable scores: This is a great question and we cannot know for sure at this point. We had some discussion on this point originally, but we have added this to the discussion why, within a dataset, the comparison of kinase activities is likely possible. We added to the discussion: “ These results also suggest that activity scores are comparable across kinases within an experiment, as the kinases with the highest activity scores in each tissue tended to be the ones you would expect, such as HCK and BTK in chronic myeloid leukemia cells and EGFR and ERBB2 in non-small cell lung carcinoma. **Since we set all kinase networks to the same size (treating tyrosine and serine/threonine kinases independently), if more phosphorylation sites were sampled from one kinase’s network then another, this directly translates to a larger activity score.**”

Minor comments:

1. The supplementary figures are a mess, with multiple figures under each figure and no legend making it difficult to follow what is going on where and what the main message is, as well as how it corresponds to the text.

Thanks for highlighting this, we have substantially improved the supplements and so hopefully it is clearer how the supplements relate to the paper. The supplements have all been changed to include significantly more structure and legends. We have added supplements as well in response to changes here.

2. Using the NetworkKIN predicted network as a background doesn’t really mitigate the literature biases that much since NetworkKIN relies on the STRING network and position specific scoring matrices for their predictions both of which suffer from bias. Perhaps this can be mentioned in the discussion since they authors stress that their method is bias-aware as a feature.

We appreciate the identification of this issue. It is true that we can identify, measure, and mitigate, but not completely overcome the underlying study bias of these networks. In our rewrite of the discussion, we highlighted this (see text changes), but also used this to point out that the general approach could likely help other algorithms. For example, we saw the same study bias we observed for LCK, translated to KSEA results.

3. The heuristic algorithm for pruning the networks could be described in a clearer way

Thanks for highlighting this. We have made substantial changes to the manuscript in this area for increased clarity and provided pseudocode in the supplement for further clarification (and to assist others should they like to reimplement the approach in a different programming language). This is a new supplemental methods file.

Reviewer #3 (Remarks to the Author):

Kinases play an important role in oncogenesis and cancer progression and are therefore often used as drug targets for cancer therapies. However, treatment challenges remain, such as the development of resistance and non-response to treatment, which could be reduced by reliable estimation of kinase activities. In this manuscript the authors present KSTAR, an algorithm that predicts kinase activity profiles from phosphoproteomic data using kinase-substrate prediction networks. It is based on the assumption that the more active a kinase is, the more of its substrates will be observed in a phosphoproteomic experiment and additionally tries to handle the kinase-substrate connections in an error- and bias-aware manner. For that, kinase-substrate prediction graphs from NetworkKIN are pruned instead of using thresholded networks to overcome high study bias, isolate less phosphorylation sites and remove high kinase network overlap. The authors validate KSTAR by predicting kinase activities from different perturbation experiments and testing the robustness of the prediction by using data across different studies and laboratories. Additionally, KSTAR was used to identify clinically-relevant kinase activity profiles using bulk cancer biopsy data. According to this

validation, KSTAR was able to reproduce expected kinase activity profiles, show robust predictions across studies from different laboratories allowing a better comparison, identify tissue-specific kinase profiles and could complement current clinical standards in the example of HER2-status identification.

The main advantage of KSTAR is that by only relying on the identification of phosphorylation sites from a sample instead of requiring relative quantification of phosphorylation sites it is able to include more phosphosites and no matched samples are required. Furthermore, it addresses study biases in the kinase-substrate network.

This could be a meaningful contribution to the field, once the following comments are adequately addressed:

Major points

1. Our main criticism is that this method is never compared to other current approaches that take the quantification of phosphorylation sites into account or use thresholded networks. It would be essential to see how well this method performs compared to others to be able to truly assess its value.

We greatly appreciate this perspective and we have significantly increased comparison. To summarize, KSTAR is the only algorithm of its kind in the field that can operate on any mass spec pipeline data to produce physiologically informative results. A very deep description can be found in response to Reviewer 2's comment of a similar nature, which we placed under the heading "Comparison to Other Approaches".

2. We think it could be useful to include further benchmarking datasets for the validation of KSTAR as in e.g. Kuleshov et al, Nucleic Acids Res., 2021, Yilmaz et al, Nature Communications, 2021, Hernandez-Armenta et al, Bioinformatics, 2017, or others.

We appreciate the suggestion and a detailed analysis and response can be found in response to Reviewer 2's comment of a similar nature and we have placed under the heading "Systematic Comparison".

3. The authors used the kinase-substrate graph of NeworKIN. However, this resource is relatively old so maybe it would be good to test the prediction using more recent resources.

We agree, it is concerning how old the original NetworkKIN is. It's what motivated us to do the background work of assembling a large variety of possibilities (and related to the last comment), sadly none of the newer algorithms provided us with anything that was physiologically relevant. Importantly, we ran NetworkKIN on the new phosphoproteome in our previous work, which means that the "old" approach is current for "new" data (i.e. the current human phosphoproteome).

We added the following summary text of observations in response to this comment and the last comment from this reviewer: "For example, PhosphoPICK-based networks failed to show ABL activity in BCR-ABL driven cancers and GPS failed to show HER2-activity in any breast cancer sample, where HER2-activity is a driving oncogene in many of these samples. Instead, GPS-based networks suggested that ERBB2 activity increases in Jurkat cells in response to TCR activation, inconsistent with tissue-specific expectations. "

Minor points

1. In the method section "KSTAR algorithm implementation and details" the last sentences of the first two paragraphs are cut off (line 466, line 473).

Thanks, our apologies. These have been corrected.

2. Authors state: "NetworkKIN results in predictions consistent with physiological expectation, but using kinase-substrate networks from other prediction algorithms (GPS and PhosphoPICK), did not result in physiologically relevant hypotheses" - could authors elaborate on this?

This is an excellent suggestion. We updated the text to summarize the key findings that demonstrated to us during testing that PhosphoPICK and GPS results were inconsistent with a minimum physiological correlation to expected tissue and conditions (that has been copied in red to major comment #3 above).

REVIEWER COMMENTS

Reviewer #3 (Remarks to the Author):

Remarks to the author

In the new manuscript the authors were able to present more clearly what is unique and novel about their method and explained their reasoning for not relying on quantification from mass spectrometry and not requiring comparative samples. They explained their heuristic pruning algorithm more clearly and improved the supplements. Additionally, they included a comparison to KEA3 and KSEA for all data sets in the manuscript and the datasets that include differential quantitative data, respectively.

These are the points that in our opinion were not fully addressed yet:

Major concerns

The main remaining concern is that a systematic evaluation of the method is still missing. Even though the authors mention valid limitations of the existing benchmarking data sets, it is still essential to see how well KSTAR is able to recover perturbed kinases and how its performance compares to other methods. At the moment, it is difficult to decide for a user why to select KSTAR over other existing methods.

Besides KSEA and KEA3 there are many other methods that estimate kinase activity as the authors also point out in their supplementary table. A comparison to these methods would be important to investigate the added value of KSTAR, especially since many of them are also able to handle single-samples (e.g. INKA, KARP).

The authors state in their discussion that “relative quantification by mass spectrometry would exacerbate the study bias...”. Here, it would be important to systematically evaluate their statement. Within KSTAR the authors present both a new network, that was pruned, and a new method, that doesn't rely on quantification from mass spectrometry. To assess both of their values separately it would be interesting to also test for example their pruned network in combination with a prediction algorithm relying on quantification from mass spectrometry (e.g. KSEA).

Minor concerns

Improve resolution of Supplementary figures 6.9 “Patient-specific KSTAR predictions for pathologically complete responders (pCR)” and 7.1 “Looking at the distribution of kinases across the dataset”.

We would like to thank the reviewer and the editor for their time and feedback.

Reviewer #3 (Remarks to the Author):

Remarks to the author

In the new manuscript the authors were able to present more clearly what is unique and novel about their method and explained their reasoning for not relying on quantification from mass spectrometry and not requiring comparative samples. They explained their heuristic pruning algorithm more clearly and improved the supplements. Additionally, they included a comparison to KEA3 and KSEA for all data sets in the manuscript and the datasets that include differential quantitative data, respectively.

These are the points that in our opinion were not fully addressed yet:

Major concerns

The main remaining concern is that a systematic evaluation of the method is still missing. Even though the authors mention valid limitations of the existing benchmarking data sets, it is still essential to see how well KSTAR is able to recover perturbed kinases and how its performance compares to other methods. At the moment, it is difficult to decide for a user why to select KSTAR over other existing methods.

Besides KSEA and KEA3 there are many other methods that estimate kinase activity as the authors also point out in their supplementary table. A comparison to these methods would be important to investigate the added value of KSTAR, especially since many of them are also able to handle single-samples (e.g. INKA, KARP).

We appreciate the feedback that further benchmarking across more methods and conditions would be helpful for readers to assess. We believe we arrived at a methodology we could be happy with that addresses some of our concerns about global benchmarking by: 1) increasing tyrosine kinase conditions of other available benchmarking datasets and 2) aggregating across multiple representations of the same kinase target to avoid overestimating or underestimating performance based on over/under representation of kinases in the benchmarking dataset. Additionally, we arrived at an experiment to benchmark a few key features of what we believe KSTAR provides – increased robustness to data loss and less sensitivity to study bias. We have added two new sections for benchmarking for accuracy and benchmarking for sensitivity to data loss and study bias. We compared KSTAR to four available algorithms (benchmarking against INKA was not feasible, given its reliance of particular MaxQuant data files, whereas we could make KARP usable on the global benchmarking dataset).

In short, we find that KSTAR is especially superior when it comes to tyrosine kinase activity prediction, likely because: 1) tyrosine kinases have been underrepresented in past benchmarking datasets and therefore these issues have been missed and 2) study bias is especially problematic in tyrosine kinase networks. KSTAR is comparable to other best-performing algorithms for serine-threonine networks, but significantly more robust to data loss and less sensitive to study bias than other algorithms in both tyrosine and serine/threonine kinase activity prediction.

The authors state in their discussion that “relative quantification by mass spectrometry would exacerbate the study bias...”. Here, it would be important to systematically evaluate their statement. Within KSTAR the authors present both a new network, that was pruned, and a new method, that doesn't rely on quantification from mass spectrometry. To assess both of their values separately it would be interesting to also test for example their pruned network in combination with a prediction algorithm relying on quantification from mass spectrometry (e.g. KSEA).

To this point, we added the power of the benchmarking set to look at whether quantification in its own right also shows study bias, which it does – sites that are more likely to undergo large fold-changes measured across a wide variety of independent systems are indeed more likely to be annotated in more compendia (last panel of Supplemental Figure 2). Additionally, we demonstrate that quantification-dependent algorithms are prone to study bias, based on rapid degradation in their performance when phosphorylation sites that are well studied are removed, relative to the same number of random sites being removed. We tie these two observations in Discussion (which, is a discussion and not results) to the possibility that quantification-dependent algorithms would be hit in this regard as well as by their underlying networks. However, we don't feel that an experiment in KSEA of our networks is within reason for this revision and that is because: 1) we don't produce one network, KSTAR is founded on the principal that any one network contains errors, and that is why it aggregates across 50 possible heuristic networks – this would require a complete redesign of other algorithms to enable this comparison and entirely new research direction and 2) we also demonstrated that KSTAR's bias aware capability is deeply dependent on the generation of decoy experiments, randomly sampled to match the study bias of the real experiment and we have not found a way to build a corollary for an experiment whose phosphorylation site is also tied to quantitative data (e.g. how do you adequately and robustly randomize selection of sites and their quantification values?). We hope that the reviewer agrees that the new data on fold-change and study bias, along with our benchmarking of sensitivity to study bias, is sufficient for a Discussion point that relative quantification likely contributes to additional study bias.

Minor concerns

Improve resolution of Supplementary figures 6.9 “Patient-specific KSTAR predictions for pathologically complete responders (pCR)” and 7.1 “Looking at the distribution of kinases across the dataset”.

Thank you for identifying these issues in the supplementary figures, we have corrected 6.9 and Supplementary Figure 7 was removed in place of a new Supplementary Figure 4 (with high resolution for the similar charts of the benchmark experiments used in this study).

REVIEWER COMMENTS

Reviewer #3 (Remarks to the Author):

The authors added a benchmark for accuracy and sensitivity to data loss and study bias which showed that KSTAR performed comparable to other best-performing methods and is more robust to data loss and study bias.

While we appreciate their work and effort and this partially addresses our concerns, our points were not fully addressed in our opinion:

We still think a separate comparison between the network pruning and the algorithm itself would be necessary, to assess each added value to the prediction individually.

The authors state in their rebuttal "...we don't produce one network, KSTAR is founded on the principal that any one network contains errors, and that is why it aggregates across 50 possible heuristic networks – this would require a complete redesign of other algorithms to enable this comparison and entirely new research direction...". Even though the authors don't produce one network, it is still unclear what effect the pruning of the network has on the results, especially considered separately from the prediction algorithm. We are uncertain why testing the pruned network with a quantification algorithm such as KSEA would require a redesign of an algorithm, it seems to only slightly modify the KSEA core function to work with a different network. We would suggest doing the network pruning the same way it is done in KSTAR and use each individual network to estimate kinase activity scores with another algorithm and then check the accuracy of the mean scores across all possible heuristic networks.

The other way around the KSTAR algorithm should be run with an unpruned network and the overrepresentation results can then be compared with the averaged overrepresentation result of the heuristic networks.

We would like to thank the reviewer and the editor for their time and feedback.

Reviewer #3:

We still think a separate comparison between the network pruning and the algorithm itself would be necessary, to assess each added value to the prediction individually.

The authors state in their rebuttal "...we don't produce one network, KSTAR is founded on the principal that any one network contains errors, and that is why it aggregates across 50 possible heuristic networks – this would require a complete redesign of other algorithms to enable this comparison and entirely new research direction...". Even though the authors don't produce one network, it is still unclear what effect the pruning of the network has on the results, especially considered separately from the prediction algorithm. We are uncertain why testing the pruned network with a quantification algorithm such as KSEA would require a redesign of an algorithm, it seems to only slightly modify the KSEA core function to work with a different network. We would suggest doing the network pruning the same way it is done in KSTAR and use each individual network to estimate kinase activity scores with another algorithm and then check the accuracy of the mean scores across all possible heuristic networks.

The other way around the KSTAR algorithm should be run with an unpruned network and the overrepresentation results can then be compared with the averaged overrepresentation result of the heuristic networks.

We appreciate the suggestion and we performed the suggested experiment, replacing the underlying information in KSEA with individual KSTAR networks and then taking the median result across the networks. These results have been added to the benchmarking section and as a supplemental figure in SFig4. The results are reported on Page 11, last paragraph and copied here:

"To assess whether the performance of other algorithms could be improved by expanding the number of kinase-substrate connections with the pruned networks developed in KSTAR, we applied KSEA to benchmarking dataset using the pruned kinase-substrate networks generated for KSTAR (Supplementary Fig. 4.6). We found that rank-based performance of KSEA improved for tyrosine kinases with the use of pruned networks, with marginal improvements for serine/threonine kinases. These gains were particularly evident when aggregating information across all 50 networks using the median activity scores, suggesting that pruned networks could potentially improve the performance of other algorithms. However, poor significance-based performance was observed, highlighting the value of generating the random null distribution used in KSTAR to improve statistical robustness of predictions. While reformulation of other algorithms like KSEA for use in the KSTAR framework is intriguing, the use of quantification in these algorithms make the generation of a random null distribution that correctly reflects the study bias and quantification distribution of real experiments considerably more difficult and beyond the scope of this work."